# CONCEPTPRUNE: CONCEPT EDITING IN DIFFUSION MODELS VIA SKILLED NEURON PRUNING

**Ruchika Chavhan**[1,2]**, Da Li**[2]**, Timothy Hospedales**[1,2]
[1]University of Edinburgh, [2]Samsung AI Center, Cambridge
Correspondence: `ruchika.chavhan@ed.ac.uk`

## ABSTRACT

While large-scale text-to-image diffusion models have demonstrated impressive image-generation capabilities, there are significant concerns about their potential misuse for generating unsafe content, violating copyright, and perpetuating societal biases. Recently, the text-to-image generation community has begun addressing these concerns by editing or unlearning undesired concepts from pre-trained models. However, these methods often involve data-intensive and inefficient fine-tuning or utilize various forms of token remapping, rendering them susceptible to adversarial jailbreaks. In this paper, we present a simple and effective training-free approach, *ConceptPrune*, wherein we first identify critical regions within pre-trained models responsible for generating undesirable concepts, thereby facilitating straightforward concept unlearning via weight pruning. Experiments across a range of concepts including artistic styles, nudity, and object erasure demonstrate that target concepts can be efficiently erased by pruning a tiny fraction, approximately 0.12% of total weights, enabling multi-concept erasure and robustness against various white-box and black-box adversarial attacks. Code available at https://github.com/ruchikachavhan/concept-prune

## 1 INTRODUCTION

In recent years, text-to-image generation has witnessed significant advances driven by the development and adoption of diffusion models (DMs) [Ho et al., 2020; Rombach et al., 2021; Ruiz et al., 2022; Saharia et al., 2022; Nichol et al., 2021; Zhang et al., 2023c; Luo et al., 2023; Podell et al., 2023] across industries and real-world scenarios. However, this swift advancement presents a substantial risk. Diffusion models can threaten artists' livelihoods through style replication [Sarah Andersen, 2023], generate convincing deepfakes and NSFW content [Review, 2023; Forensics, 2024], and perpetuate societal biases [Luccioni et al., 2023]. The risks associated with large-scale text-to-image models arise from billion-sized web-scraped datasets used in training, comprising public datasets like LAION [Schuhmann et al., 2022], COYO [Byeon et al., 2022], and CC12M [Changpinyo et al., 2021], that often lack human-level quality assurance. A simplistic and naive solution to mitigate these risks involves fine-tuning the model on datasets without this undesired content; however, this approach can prove to be highly compute-expensive.

Several efforts addressing the risks of diffusion models have been made from the perspective of Concept Editing [Kumari et al., 2023; Gandikota et al., 2023a;b; Zhang et al., 2023a; Orgad et al., 2023] and Model Unlearning (MU) [Heng & Soh, 2023; Zhao et al., 2024; Liu et al., 2024; Wu et al., 2024; Fan et al., 2023], both aimed at eliminating undesired prompts, albeit with differing objectives. Concept editing methods seek to eliminate undesired prompts by aligning latent representations of the target concept with a concept to be retained, via methods such as maximizing similarity [Kumari et al., 2023; Gandikota et al., 2023a] and token remapping [Zhang et al., 2023a; Gandikota et al., 2023b]. Conversely, Model Unlearning formulates an objective that penalizes forgetting desired concepts while promoting the elimination of undesired ones, but this requires expensive computations and fine-tuning. Moreover, as most concept editing approaches rely on some form of token blacklisting or resteering [Zhang et al., 2023a], adversarial attacks based on textual inversion [Zhang et al., 2023d; Pham et al., 2023; Yang et al., 2023; Tsai et al., 2024] have demonstrated the ability to circumvent concept erasure methods [Gandikota et al., 2023a;b; Zhang et al., 2023a] that were previously believed to be robust with a near-perfect success rate.

In this paper, we introduce *ConceptPrune*, an entirely training-free method for concept editing that, for the first time, tackles knowledge editing in diffusion models through the lens of pruning. Leveraging recently introduced pruning heuristics [Sun et al., 2024], we identify regions or neurons in feed-forward layers of diffusion models that strongly activate in the presence of a concept, and denote them as *skilled neurons*. Subsequently, concept removal can be achieved by simply pruning or *zeroing* out these skilled regions. We demonstrate that ConceptPrune provides a rapid, efficient, and unified solution for erasing undesired concepts, including various artist styles, nudity, undesired objects, and gender biases. Notably, it maintains the outstanding image-generation prowess of pre-trained models while remaining resilient to adversarial attacks.

## 2 RELATED WORK

**Concept Erasure in Diffusion Models:** Concept erasure has gained significant attention and has rapidly emerged as a pivotal area of research in diffusion models. Recent concept erasure methods can be broadly categorized into two main areas: *Model Unlearning* and *Concept Editing*.

Model Unlearning methods [Heng & Soh, 2023; Wu & Harandi, 2024; Fan et al., 2023; Zhang et al., 2024] typically require extensive training to forget a target concept while preserving unrelated ones. While these methods have shown remarkable efficacy in unlearning multiple concepts, they are usually computationally expensive, especially for large-scale models.

Concept Editing [Gandikota et al., 2023a;b; Kumari et al., 2023; Zhang et al., 2023a; Huang et al., 2023; Lu et al., 2024; Lyu et al., 2023] focuses on modifications to specific parts of the model. These edits ensure that the denoised output for the target concept aligns with clean, desired concepts. The training costs associated with Concept Editing can be mitigated by strategies such as tuning only cross-attention weights [Gandikota et al., 2023a; Kumari et al., 2023; Zhang et al., 2023a; Huang et al., 2023; Lu et al., 2024], solving closed-form objectives to update attention parameters[Gandikota et al., 2023b; Lu et al., 2024; Orgad et al., 2023], or parameter-efficient adaptation like LORA [Hu et al., 2022] to edit the model [Lu et al., 2024; Lyu et al., 2023].

While the aforementioned methods are highly effective, deploying current state-of-the-art concept erasure techniques in real-world scenarios poses significant challenges, particularly in online environments with computational constraints where harmful concepts can emerge dynamically. This is because these methods struggle to meet the following requirements for real-world applications: (1) *training-free concept erasure*, eliminating concepts without the need for backpropagation through the entire model, or (2) *lightweight or fast concept erasure*, allowing concepts to be removed quickly and efficiently with minimal compute.

Most concept-erasure methods rely on extensive fine-tuning and are therefore not training-free, however some training-based approaches like UCE [Gandikota et al., 2023b], SPM [Lyu et al., 2023], MACE [Lyu et al., 2023], and FMN [Zhang et al., 2023a] are notably *lightweight* and suitable for the real-world setting. For instance, FMN erases concepts in 30 seconds and UCE in about 2 minutes, while the rapid fine-tuning of LoRA parameters makes SPM and MACE ideal for real-time online erasure. In Table 1, we present a comprehensive summary of related works, categorizing them based on whether they are training-free and lightweight for an online setting.

Our proposed solution, ConceptPrune, excels on both fronts by introducing a training-free, pruning-based approach that eliminates harmful concepts without updating any parameters. Instead, it identifies and targets the neurons responsible for generating these concepts enabling efficient concept erasure with significantly reduced computational requirements.

**Language model skilled neuron identification:** Previous works [Wang et al., 2022; Suau et al., 2020; Durrani et al., 2023; Dalvi et al., 2018; Durrani et al., 2020; Antverg & Belinkov, 2022] present strong evidence that activation of specific neurons in feed-forward networks in transformers show high correlation with task labels, with perturbations to these neurons impacting task performance. Modular components within pre-trained transformers were identified by leveraging the inherent sparsity in neurons, as shown in [Zhang et al., 2022]. Further, [Zhang et al., 2023e] demonstrates that these modules are specialized in distinct functions. In this work, we aim to identify neurons accountable for generating undesired concepts in diffusion models — a pursuit hitherto unexplored in this domain. Unlike language models, identifying neurons in diffusion models is complicated due to

| Method | Training-free | Parameters Trained | Lightweight Erasure |
|---|---|---|---|
| CA [Kumari et al., 2023] | ✗ | Full denoiser | ✗ |
| SA [Heng & Soh, 2023] | ✗ | Full denoiser | ✗ |
| SH [Wu & Harandi, 2024] | ✗ | Full denoiser | ✗ |
| AdvUnlearn [Zhang et al., 2024] | ✗ | Full denoiser | ✗ |
| SalUn [Fan et al., 2023] | ✗ | Full denoiser | ✗ |
| ESD [Gandikota et al., 2023a] | ✗ | Cross Attention | ✗ |
| Receler [Huang et al., 2023] | ✗ | Cross Attention | ✓ |
| FMN [Zhang et al., 2023a] | ✗ | Cross Attention | ✓ |
| SPM [Lyu et al., 2023] | ✗ | LORA | ✓ |
| MACE [Lu et al., 2024] | ✗ | Cross Attention + LORA | ✓ |
| UCE [Gandikota et al., 2023b] | ✓ | Cross Attention | ✓ |
| Ours (ConceptPrune) | ✓ | ***None*** | ✓ |

Table 1: Summary of recent Concept Erasure baselines. ConceptPrune is a training-free approach that enables rapid pruning of the model to eliminate a new target concept without the need for extensive re-training.

the intricate aggregation of neurons across multiple denoising time steps and the model's sensitivity to the output of previous time steps.

**Language model pruning:** Network pruning [LeCun et al., 1989; Liu et al., 2019; Han et al., 2015; Frankle & Carbin, 2019; Blalock et al., 2020] aims to reduce model size either by eliminating parameters and substructures from networks [Li et al., 2017; Frantar & Alistarh, 2023] or by masking parameters guided by a score function [Frantar & Alistarh, 2023; Frantar et al., 2023; Sun et al., 2024; Lee et al., 2019]. This study primarily focuses on the latter approach. Exploration of diffusion model pruning is limited, although one study [Fang et al., 2023] introduces structural pruning by accumulating gradient-based importance scores across a chosen subset of denoising time steps. A study [Wei et al., 2024] explores safety-aligned LLMs that inhibit harmful prompts by leveraging pruning heuristics to identify regions responsible for denying harmful responses. In contrast, we use pruning heuristics to locate critical weight regions responsible for unsafe behaviors in pre-trained models and permanently unlearn them through pruning.

## 3 PRELIMINARIES

**(Latent) diffusion models:** Diffusion models (DMs) [Ho et al., 2020; Song et al., 2021] are essentially image denoisers that learn to reverse a forward Markov process in which noise is added into input images for multiple time steps $t \in [0, T]$. During training, given a real image $\mathbf{x}_0$, a noisy image $\mathbf{x}_t$ at time $t$ is obtained by $\sqrt{a_t}\mathbf{x}_0 + \sqrt{1 - a_t}\epsilon$, where $\epsilon \sim \mathcal{N}(0, I)$ and $a_t$ is a gradually decaying parameter. Then, the denoiser learns to predict the noise added for obtaining $\mathbf{x}_t$, such that $\mathbf{x}_0$ can be reconstructed back by deducting predicted noise from $\mathbf{x}_t$.

Latent diffusion models (LDMs) [Rombach et al., 2022b; Zhang et al., 2023b] are widely used as the first choice of DMs as they accelerate the above process by operating in a latent space $\mathbf{z}$, of input $\mathbf{x}$. Thus, a LDM consists of a latent embedding denoiser $f_\theta(.)$, which is trained to predict the added noise by stochastically minimizing the objective $\mathcal{L}(\mathbf{z}, p) = \mathbb{E}_{\epsilon, \mathbf{x}, p, t} [\|\epsilon - f_\theta(\mathbf{z}_t, p, t)\|]$. Given a text prompt $p$, an encoder which extracts $\mathbf{z}_0$ from $\mathbf{x}_0$ and a decoder which maps the denoised $\hat{\mathbf{z}_0}$ to the pixel space. To synthesize an image during inference based on text prompt $p$, one first samples a noisy embedding $\mathbf{z}_T$ which is iteratively denoised for $T$ time steps until $\hat{\mathbf{z}}_0$ for generating the final image is obtained. Normally, the encoder and decoder are obtained from a frozen pre-trained autoencoder.

## 4 CONCEPTPRUNE: A TRAINING-FREE CONCEPT EDITING FRAMEWORK

**Motivation:** Concept editing methods aim to eliminate the undesired concept from a pretrained DM. Inspired by the observation that concepts can activate specific neurons in a neural network [Mahendran & Vedaldi, 2015; Wang et al., 2022], we ask the question: *Can we remove an undesired concept from a pre-trained DM by simply finding neurons specific to this concept, and pruning them? The answer is yes.* We show that neurons in LDMs often specialise to specific concepts, and

that pruning these neurons can be used to permanently eliminate undesired concepts from image generation.

## 4.1 FEED FORWARD NETWORKS (FFNs) IN LATENT DIFFUSION MODELS

We focus on a pre-trained LDM, i.e. Stable Diffusion [Rombach et al., 2021], characterized by a UNet [Ronneberger et al., 2015] denoted as $f_\theta$. The UNet architecture incorporates two ResNet blocks that sandwich two transformer blocks with self-attention between latent representations, cross-attention for the transfer of information from conditional inputs to latent representations, and a Feed-forward Network (FFN) with GEGLU activation [Shazeer, 2020]. Prior research in concept editing, such as [Gandikota et al., 2023a] and [Zhang et al., 2023a], primarily examines cross-attention or self-attention visualizations to detect concept presence or generation. Diverging from this approach and drawing inspiration from NLP skill discovery [Suau et al., 2020; Wang et al., 2022; Zhang et al., 2023e; Durrani et al., 2020; Dalvi et al., 2018], our focus lies on neurons within the Feed-forward networks.

We begin by denoting the input to the FFN layer $l$ at time step $t$ for text prompt $p$ by $\mathbf{z}_t^l(p) \in \mathbb{R}^{d \times N}$, where $N$ is the number of latent tokens and corresponding output by $\mathbf{z}_t^{l+1}(p) \in \mathbb{R}^{d \times N}$. FFN in Stable Diffusion consists of GEGLU activation [Shazeer, 2020] which operates as shown in Equation 1.

$$\mathbf{h}_t^l(p) = \sigma(\mathbf{W}^{l,1} \cdot \mathbf{z}_t^l(p)) \tag{1}$$
$$\mathbf{z}_t^{l+1}(p) = \mathbf{W}^{l,2} \cdot \mathbf{h}_t^l(p)$$

where, $\mathbf{W}^{l,1} \in \mathbb{R}^{d' \times d}$, $\mathbf{W}^{l,2} \in \mathbb{R}^{d \times d'}$ are weight matrices in the first and second linear layers, bias terms are omitted for simplicity and $\sigma(\cdot)$ is GEGLU activation [Hendrycks & Gimpel, 2023]. In our work, we regard $\mathbf{W}^{l,2}[i,:]$ the $i$-th row and $\mathbf{W}^{l,2}[i,j]$ the element in $i$-th row and $j$-th column of matrix $\mathbf{W}^{l,2}$.

## 4.2 PRUNING STRATEGY: WANDA

We start with recapping the pruning method Wanda [Sun et al., 2024] for the large language models (LLMs), and its adaptation to diffusion models. We denote the weights of linear layer by $\mathbf{W} \in \mathbb{R}^{d_{out} \times d_{in}}$ and input $\mathbf{X} \in \mathbb{R}^{B \times d_{in}}$, where $B$ is the number of data points, i.e. the number of prompts in this paper. Unlike magnitude-based pruning, which considers the weights' magnitude alone, the concept behind the Wanda score is to estimate the combined effect of weights and the magnitude of features on neuron activations. Therefore, the importance of each weight is calculated as an element-wise product of its magnitude and the corresponding input feature-dimension-wise $\ell_2$ norm as shown in Equation 2

$$\mathbf{S}(\mathbf{W}, \mathbf{X}) = |\mathbf{W}| \odot \left(\mathbf{1}^{d_{out}} \cdot \|\mathbf{X}\|_2\right) \in \mathbb{R}^{d_{out} \times d_{in}}. \tag{2}$$

Here $|\cdot|$ to denote the absolute value operator, $\|\mathbf{X}\|_2$ computes the $\ell_2$ norm of each column of $\mathbf{X}$ and results in a $d_{in}$ dimensional vector, and $\odot$ represents element-wise matrix multiplication. Specifically, Eq 2 broadcasts $\|\mathbf{X}\|_2$ across different rows of $\mathbf{W}$ for computing the element-wise product in each row. For each row of $\mathbf{W}$, represented by $\mathbf{W}_{i,:}$ with corresponding Wanda score $\mathbf{S}(\mathbf{W}, \mathbf{X})_{i,:}$, the bottom-$k\%$ weights with the lowest scores are zeroed out [Sun et al., 2024]. This process effectively induces sparsity in each row of the weights $\mathbf{W}$ by eliminating the bottom-$k\%$ of the weights, as a row is connected to a single activation in the output of a linear layer as a *per-output basis* [Sun et al., 2024]. Elements of the weight matrix $\mathbf{W}$ are often referred to as *weight neurons*, which are different from neurons corresponding to the output of a layer. After pruning the least important weight neurons in a layer, subsequent layers in the model receive updated input activations. Wanda does not require any costly weight update since it solely relies on a calibration set to compute the feature norm matrix, which can be obtained with just a single forward pass through the model. The following will discuss how we use Wanda to prune each row's top-$k\%$ weight neurons for eliminating a concept.

## 4.3 IDENTIFYING SKILLED NEURONS IN LATENT DIFFUSION MODELS

**Target and reference concept prompts:** We first define two sets of calibration prompts $\mathcal{P}^* = \{p_1^*, p_2^*, ..., p_M^*\}$ and $\mathcal{P} = \{p_1, p_2, ...p_M\}$ using $M$ objects that can be generated by the model in target and reference concepts, respectively. Here, $p_i^*$ and $p_i$ represent prompts with the target and

reference concepts, respectively. Objects represent common categories, including 'cat', 'dog', etc. To eradicate the target concept, e.g., "Van Gogh" painting style, we formulate a $p_i^*$ as `a <object> in Van Gogh style'` and a $p_i$ as `a <object>'`.

**Importance score for FFN weights at time $t$:** We begin by collecting the neuron activations described in Eq 1, corresponding to the sets of target concept and reference prompts, and shape them into matrices denoted by $\mathbf{H}_t^l(\mathcal{P}^*) = [\mathbf{h}_t^l(p_1^*)^T, \mathbf{h}_t^l(p_2^*)^T, ..., \mathbf{h}_t^l(p_M^*)^T]$ and $\mathbf{H}_t^l(\mathcal{P}) = [\mathbf{h}_t^l(p_1)^T, \mathbf{h}_t^l(p_2)^T, ..., \mathbf{h}_t^l(p_M)^T]$ such that $\mathbf{H}_t^l(\mathcal{P}^*), \mathbf{H}_t^l(\mathcal{P}) \in \mathbf{R}^{(M*N) \times d'}$. Note that this process only requires one forward pass for per prompt.

After collecting both sets of neuron activations, we calculate the importance score for the linear weight $\mathbf{W}^{l,2}$ in Eq 1 for both target and reference prompts using the methodology described in 4.2 and Eq 2 as

$$\mathbf{S}(\mathbf{W}^{l,2}, \mathbf{H}_t^l(\mathcal{P}^*)) = |\mathbf{W}^{l,2}| \odot (\mathbf{1}^d \cdot \|\mathbf{H}_t^l(\mathcal{P}^*)\|_2) \tag{3}$$
$$\mathbf{S}(\mathbf{W}^{l,2}, \mathbf{H}_t^l(\mathcal{P})) = |\mathbf{W}^{l,2}| \odot (\mathbf{1}^d \cdot \|\mathbf{H}_t^l(\mathcal{P})\|_2)$$

For ease of notation, we denote $\mathbf{S}(\mathbf{W}^{l,2}, \mathbf{H}_t^l(\mathcal{P}^*))$ and $\mathbf{S}(\mathbf{W}^{l,2}, \mathbf{H}_t^l(\mathcal{P}))$ as $\mathbf{S}_t^l(\mathcal{P}^*)$ and $\mathbf{S}_t^l(\mathcal{P})$ respectively in the subsequent sections. Following this, we identify a skilled neuron by comparing its importance score for the target concept prompt with that for the reference prompt.

**Isolating concept-generating neurons at time $t$:** Similar to Wanda [Sun et al., 2024], we adopt a *per-output comparison group*, which considers the importance scores among weights in each row of the weight matrix, rather than the matrix as a whole. Specifically, for a given sparsity level $k\%$, we define the top-$k\%$ important weight neurons for generating the target concept in row-$i$ denoted by $\mathbf{W}^{l,2}[i,:]$ as

$$\mathbf{I}_t^l(\mathcal{P}^*)[i,j] = \begin{cases} 1 & \text{if } \mathbf{S}_t^l(\mathcal{P}^*)[i,j] \in \text{top-}k\% \text{ of } \mathbf{S}_t^l(\mathcal{P}^*)[i,:] \\ 0 & \text{otherwise,} \end{cases} \tag{4}$$

where $\mathbf{I}_t^l(\mathcal{P}^*)$ forms a binary mask matrix for the concept prompt set $\mathcal{P}^*$. As $\mathcal{P}^*$ contains additional undesired target concepts compared with $\mathcal{P}$, $\mathbf{I}_t^l(\mathcal{P}^*)$ thus consists of the set of important neurons that are responsible for generating both the target and reference concepts. Our next step involves filtering and disentangling these neurons to isolate them to generate the target concept and the reference separately. Continuing with comparison on the Wanda score matrices for both target and reference prompts sets, we now define *skilled* neurons.

**Definition 4.1** *For a linear layer characterized by $\mathbf{W}^{l,2}$, the weight neuron $\mathbf{W}^{l,2}[i,j]$ is defined as a **skilled** neuron at time step $t$ if $\mathbf{I}_t^l[i,j](\mathcal{P}^*) == 1$ and $\mathbf{S}_t^l(\mathcal{P}^*)[i,j] > \mathbf{S}_t^l(\mathcal{P})[i,j]$.*

In essence, if a weight neuron ranks within the top-$k\%$ Wanda scores among other neurons in a row of $\mathbf{W}^{l,2}$ for the target prompts $\mathcal{P}^*$, it contributes to generating either the undesired target concept or the reference concept. However, if its Wanda score surpasses that of a reference concept, it predominantly influences the target concept.

Subsequently, we form a time-dependent binary mask $\mathbf{M}_t^l$ over weight matrix $\mathbf{W}^{l,2}$ such that

$$\mathbf{M}_t^l[i,j] = \begin{cases} 1 & \text{if weight neuron } \mathbf{W}^{l,2}[i,j] \text{ is skilled 4.1} \\ 0 & \text{otherwise,} \end{cases} \tag{5}$$

where $\mathbf{M}_t^l$ is a subset of $\mathbf{I}_t^l$ as only neurons that are highly activated by the target concept are retained.

**Removing aggregated skilled neurons over timesteps:** While we previously described time-dependent skilled neurons, DiffPrune [Fang et al., 2023] demonstrates that weights can be pruned by aggregating a pruning metric over a selected subset of timesteps based on relative importance scores. However, in our study, we discovered that simply aggregating the binary mask over the first $\hat{t}$ denoising iterations suffices to eliminate a concept while preserving the underlying object. Consequently, we define pruned weight matrix $\hat{\mathbf{W}}^{l,2}$ as

$$\hat{\mathbf{W}}^{l,2} = \mathbf{W}^{l,2} \odot (\neg(\vee_{t=T,T-1,...,T-\hat{t}} \mathbf{M}_t^l)) \tag{6}$$

where $\vee$ and $\neg$ denote the logical OR and NOT operators. All the weights of the pre-trained diffusion model $f_\theta$ remain unchanged as only $\mathbf{W}^{l,2}$ is substituted with pruned weights obtained from Equation 6. We then perform experiments with the pruned model to evaluate the effectiveness of concept removal, i.e. subsequently, we only use $\hat{\mathbf{W}}^{l,2}$ for image sampling.

## 5 EXPERIMENTS

### 5.1 EXPERIMENT DETAILS

We work with Stable Diffusion v1.5 (SD), which includes 16 FFN layers that serve as candidates for skilled neuron discovery and pruning. We begin by formulating the calibration sets $\mathcal{P}^*$ and $\mathcal{P}$ that are used to obtain the matrices $\mathbf{H}_t^l(\mathcal{P}^*)$ and $\mathbf{H}_t^l(\mathcal{P})$ for calculating the score in Equation 3. The list of prompts and the exact structure of the sentences for different concepts is provided in Table 9 in the Appendix.

**Pruning candidates:** The selection of FFN second layer for pruning was informed by an ablation study we conducted across various layers within the UNet, aimed at identifying the most effective pruning targets. Specifically, we analyzed the first layer of the FFNs, along with the query, key, and value weight matrices in all cross-attention layers. In Appendix A.2, we present the concept erasure performance for pruning within these layers, as well as an analysis of neuron activation patterns. The results clearly indicate that the second layer of the FFNs proves to be a more effective pruning candidate compared to other layers. This observation aligns with findings in the LLM literature, where these layers have also been identified as prime candidates for skill discovery and pruning [Zhang et al., 2023e; Suau et al., 2020; Wang et al., 2022]. Finally, to calculate neuron activations, we run the model for 50 denoising iterations and fix the seed before every forward pass to ensure the same initializations for both reference and target concept prompts.

**Hyper-parameter selection:** As discussed in Section 4.1, we select two key hyperparameters—sparsity level $k\%$ and $\hat{t}$—for aggregating skilled neurons across time steps. For each concept, we vary the sparsity parameter $k\%$ between $0.5\%$ and $5\%$, choosing the value that achieves the best trade-off between concept erasure and the retention of unrelated concepts. More details on this hyperparameter selection process can be found in Section A.3 of the appendix. The optimal sparsity levels $k\%$ and the corresponding $\hat{t}$ values for each concept are outlined in Table 10 in the appendix. Interestingly, our experiments reveal that $\hat{t} = 10$ is typically sufficient for erasing concepts while preserving objects, suggesting that low-level features such as style and objects are formed early in the denoising process, with fine-grained details being added later.

**Baselines:** We identify the following concept editing methods as our closest competitors due to their lightweight approach: UCE[Gandikota et al., 2023b], Forget-Me-Not (FMN) [Zhang et al., 2023a], MACE [Lu et al., 2024], Receler [Huang et al., 2023], and SPM [Lyu et al., 2023]. These works are considered direct competitors as they share a similar emphasis on computational efficiency. Additionally, we include training-intensive methods such as Concept Ablation (CA) [Kumari et al., 2023], ESD [Gandikota et al., 2023a], Selective Amnesia (SA)[Heng & Soh, 2023], Scissorhands (SH)[Wu & Harandi, 2024], and AdvUnlearn [Zhang et al., 2024] in our comparison. However, we categorize these as indirect competitors, as their reliance on extensive fine-tuning contrasts with ConceptPrune's training-free regime. We include a baseline only if their method has been evaluated for that concept and is reproducible from their source code.[1]

### 5.2 ERASING ARTISTIC STYLES

We consider five artists — *Van Gogh*, *Claude Monet*, *Pablo Picasso*, *Leonardo Da Vinci*, and *Salvador Dali*. To measure the efficacy of concept removal, we created a dataset of 50 prompts for each artist using ChatGPT, consisting of the names of their paintings followed by the name of the artist. To measure the efficacy of concept removal, we report two metrics: the *CLIP Similarity*, which measures the similarity between the generated image and the prompt, and a stricter *CLIP score* that penalizes a model when the similarity between the image generated by the concept-editing and the prompt is greater than the similarity between the image generated by the pre-trained SD and prompt. Lower values of *CLIP Similarity* and higher values of *CLIP Score* indicate better concept removal. We also evaluate the fidelity of general purpose image generation by measuring FID and *CLIP Similarity* on the COCO30k dataset. From the quantitative results presented in Table 2, we demonstrate that our method outperforms other baselines in artist style removal while effectively retaining unrelated concepts, as indicated by the low FID score. In Figure 1, we present some qualitative examples that demonstrate the strong erasing capabilities of ConceptPrune with high-quality realistic output

---

[1]We reproduced CA to remove nudity and object classes from ImageNette but performance was very poor.

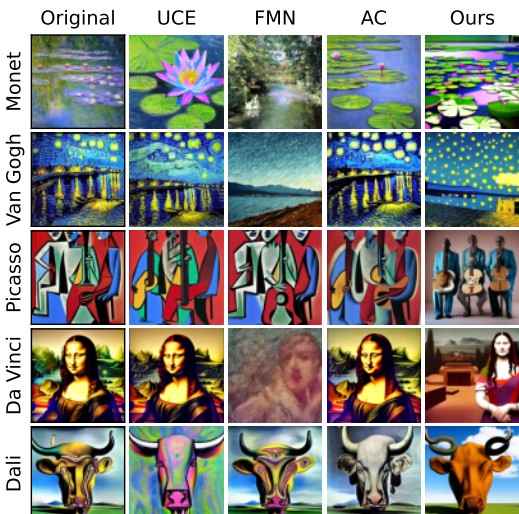

Figure 1: Qualitative results of artist erasure. ConceptPrune demonstrates stronger erasing while generating high-quality, realistic-looking images.

Table 2: Quantitative results of Artist style removal, average over 5 artist styles. CLIP Similarity and CLIP Accuracy measure art style removal. FID and CLIP Similarity on COCO30k measure fidelity for unrelated retained concepts. The full split of the results for different art styles is reported in the appendix in Table 11. Our ConceptPrune can effectively erase artist styles without compromising the model's performance on unrelated concepts.

| Light-weight | | Artist erasure | | COCO | |
| --- | --- | --- | --- | --- | --- |
| | | Similarity ↓ | Score ↑ | FID ↓ | Similarity ↑ |
| | Original SD | 42.1 | 23.0 | **14.5** | **31.3** |
| ✗ | ESD | 34.1 | 49.2 | 15.9 | 30.7 |
| | CA | 32.4 | 65.2 | 17.5 | 31.3 |
| | SA | 27.1 | 86.9 | 14.7 | 31.3 |
| | AdvUnlearn | 27.2 | 82.0 | 16.9 | 29.7 |
| ✓ | UCE | 32.8 | 44.0 | 15.7 | **31.3** |
| | FMN | 28.4 | 82.4 | 20.9 | 29.8 |
| | MACE | 28.2 | 85.4 | 15.1 | 31.0 |
| | Receler | 28.4 | 82.0 | 16.7 | 29.1 |
| ✓ | Ours | **26.9** | **94.0** | 16.9 | 29.9 |

images. More qualitative results are presented in Section A.4 in the appendix. While we demonstrate strong retention of unrelated concepts in COCO30k in Table 2, Section A.4 in the appendix further provides evidence that using ConceptPrune to erase an artist's style results in minimal degradation when generating other similar artist styles.

## 5.3 ERASING EXPLICIT CONTENT

We quantitatively evaluate our proposed method for moderating Not-Safe-for-Work (NSFW) concepts like nudity by comparing it against the concept-erasing baselines ESD, UCE, and FMN. In addition, we also compare with variants of Stable Diffusion, such as Safe Latent Diffusion (SLD) [Schramowski et al., 2023] and Stable Diffusion 2.0 [Rombach et al., 2022a], which have been fine-tuned on a filtered subset of LAION without explicit images. We use the Inappropriate Prompts Dataset (I2P) [Schramowski et al., 2023], which consists of 4703 prompts featuring various inappropriate concepts. Nudity detectors [Bedapudi, 2022] indicate that, out of these 4703 prompts, the pre-trained Stable Diffusion model generates nudity for 796 prompts. In Figure 2, we report the percentage reduction in the number of generated images with nudity compared to the pre-trained Stable Diffusion model. ConceptPrune generates nudity in merely 47 prompts within 4703 prompts in the I2P dataset, implying a 94.1% decrease compared to 88% in ESD and 85.6% in UCE, demonstrating a significant improvement over other baselines in content moderation. We present more qualitative results on the I2P dataset in Figure 13 in the appendix.

## 5.4 ERASING OBJECTS

**Single-object erasing:** We showcase the effectiveness of our method in removing objects from the learned concepts of diffusion models. We conducted experiments targeting ImageNette classes [Howard & Gugger, 2020], a subset of ImageNet [Deng et al., 2009] comprising 10 classes. Similar to UCE and ESD, we generated 500 images per class and evaluated the top-1 classification accuracy using a pre-trained ResNet-50 [He et al., 2015]. Table 4 shows that ConceptPrune has superior erasure performance on average while effectively minimizing interference on non-targeted classes. [2] More results of object erasure are provided in Figure 14 in the appendix.

---

[2]We copied the numbers from their original papers based on SD 1.4 and therefore, we repeated our experiments with SD 1.4 for consistency.

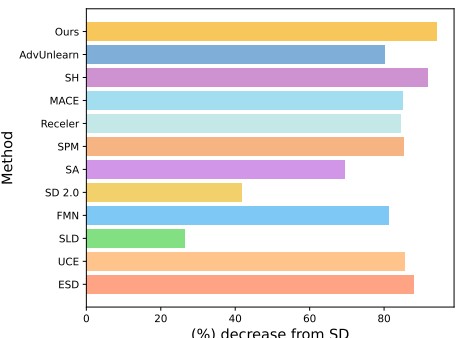

Figure 2: Explicit Content Erasure. The percentage reduction in nudity content from I2P prompts, compared to the original SD model ConceptPrune (SD1.5) decreases the number of explicit images by 94.1%, outperforming competitors as well as SD2.0.

Table 3: ConceptPrune demonstrates robustness to adversarial attacks. Unlearning methods evaluated against three adversarial attacks. Black-box (Ring-A-Bell[Tsai et al., 2024], and MMA[Yang et al., 2023]) performance is quantified by percentage reduction in nude samples compared to SD. White-box UnlearnDiffAtk [Zhang et al., 2023d] performance measures the attack success rate (ASR).

| Light-weight | Method | Ring-A-Bell ↑ | MMA ↑ | UnlearnDiffAtk ↓ |
|---|---|---|---|---|
| − | SLD | 2.8 | 25.5 | 82.4 |
| | SDv2 | 1.8 | 26.8 | 73.8 |
| ✗ | ESD | 52.8 | 87.3 | 76.1 |
| | SA | 84.3 | 94.3 | **11.3** |
| | SH | **86.1** | 94.3 | 22.3 |
| | AdvUnlearn | 85.8 | 93.7 | 21.1 |
| ✓ | UCE | 67.6 | 63.3 | 93.2 |
| | Receler | 67.9 | 65.7 | 92.1 |
| | MACE | 56.4 | 57.9 | 89.3 |
| | FMN | 5.6 | 53.6 | 97.9 |
| | SPM | 34.5 | 78.4 | 91.6 |
| ✓ | Ours | 85.2 | **95.6** | 64.8 |

Table 4: Concept Erasure: Top-1 classification accuracy of erased and preserved class samples, using a pre-trained ResNet-50. Our ConceptPrune effectively erases objects from pre-trained models without impacting the accuracy for other object classes.

| Classes | Accuracy of Erased Classes ↓ | | | | Accuracy of Preserved Classes ↑ | | | |
|---|---|---|---|---|---|---|---|---|
| | ESD | UCE | FMN | ConceptPrune | ESD | UCE | FMN | ConceptPrune |
| Church | 54.2 | 8.4 | 2.0 | **6.0** | 80.2 | 77.5 | 57.8 | **82.8** |
| English Springer | 6.2 | 0.2 | 1.9 | **0.0** | 62.6 | 78.9 | 73.5 | **80.1** |
| Golf ball | 5.8 | 0.8 | 13.7 | **0.0** | 65.6 | 79.0 | 82.8 | **87.8** |
| Gas Pump | 8.6 | **0.0** | 7.9 | **0.0** | 66.5 | **80.7** | 79.0 | 83.0 |
| Tench | 9.6 | **0.0** | 5.7 | **0.0** | 66.6 | 79.3 | 78.4 | **85.0** |
| Parachute | 23.8 | **1.4** | 8.3 | 7.0 | 65.4 | 77.4 | 98.2 | 80.6 |
| Cassette Player | 0.6 | **0.0** | 1.0 | 1.0 | 64.5 | 90.3 | 68.7 | 94.3 |
| Chain Saw | 6.0 | **0.0** | 0.1 | **0.0** | 71.6 | 80.2 | 78.4 | **91.5** |
| French Horn | 0.4 | 3.0 | **0.0** | 3.0 | 77.0 | 80.1 | 78.3 | **88.2** |
| Garbage Truck | 10.4 | 14.8 | 0.1 | **0.0** | 51.5 | **78.7** | 74.9 | 85.8 |
| Average | 12.5 | 2.7 | 4.1 | **1.7** | 66.9 | **80.2** | 77.5 | **85.9** |

**Multi-object erasing:** In addition to single-object erasing, we also evaluate ConceptPrune on removing multiple objects from the model simultaneously. Although our pruning strategy generates a pruning mask for concepts individually, it provides a straightforward baseline for multi-object erasing by taking the union of skilled neurons across different concepts. We direct the reader to Appendix A.5 for more details. We compare our method with UCE and report the accuracy on erased classes along with FID and CLIP similarity on COCO30k. Table 6 shows that ConceptPrune demonstrates comparable erasing performance while excelling at retaining unrelated concepts.

## 5.5 ADVERSARIAL DEFENSE ON CONCEPT ERASURE ATTACKS

**White-box attacks:** Recent research has recognized the limitations of the concept editing baselines considered in this paper, namely UCE, ESD, FMN, and CA. Model-based adversarial attacks like UnlearnDiffAtk [Zhang et al., 2023d] and Concept Inversion (CI) [Pham et al., 2023] have demonstrated that subtle perturbations to text prompts can circumvent the unlearning mechanisms, compelling concept-editing baselines to generate harmful images with undesired concepts once again. Furthermore, these studies show a near-perfect Attack Success Rate (ASR) for FMN and UCE which jeopardizes the safety and effectiveness of these baselines in real-world settings.

Table 5: ConceptPrune is substantially more robust to adversarial attacks aimed at eliciting erased concepts. **(Top):** Attack Success Ratio (ASR %, ↓) of UnlearnDiffAtk [Zhang et al., 2023d] adversarial prompts for Van Gogh's painting style and 4 classes of the Imagenette dataset.

| Light-weight | | Artist Style | | Object erasing | | | |
|---|---|---|---|---|---|---|---|
| | | Vincent Van Gogh | | Parachute | Tench | Garbage Truck | Church |
| | | Top-1 ASR | Top-3 ASR | ASR | ASR | ASR | ASR |
| ✗ | ESD | 32.0 | 76.0 | 54.0 | 36.0 | 24.0 | 60.0 |
| | CA | 77.0 | 92.0 | – | – | – | – |
| | SH | – | – | 24.0 | 8.0 | 2.0 | **6.0** |
| | SalUn | – | – | 74.0 | 14.0 | 42.0 | 62.0 |
| | AdvUnlearn | **2.0** | **24.6** | **14.0** | **4.0** | 8.0 | **6.0** |
| ✓ | UCE | 94.0 | 100.0 | 43.0 | 22.0 | 38.0 | 68.0 |
| | FMN | 56.0 | 90.0 | 100.0 | 100.0 | 98.0 | 96.0 |
| | SPM | – | – | 96.0 | 90.0 | 82.0 | 94.0 |
| ✓ | ConceptPrune (Ours) | **2.4** | **24.4** | 34.0 | 16.1 | **0.0** | 21.7 |

Table 6: Quantitative results for multi-object erasure. We report Accuracy on erased classes and FID on COCO30k, CLIP similarity on COCO30k, and ASR of UnlearnDiffAtk. ConceptPrune is comparable to UCE at erasing multiple objects and outperforms UCE in retaining image generation capabilities along with being significantly robust to white-box adversaries.

| | COCO FID | CLIP score | Accuracy on erased classes | ASR |
|---|---|---|---|---|
| UCE | 17.7 | 31.0 | 4% | 22% |
| ConceptPrune | 17.5 | 29.9 | 7% | 6% |

We evaluate ConceptPrune under these recently introduced white-box attacks - UnlearnDiffAtk [Zhang et al., 2023d] and Concept Inversion (CI) [Pham et al., 2023]. For UnlearnDiffAtk, we evaluate for Van Gogh style, ImageNette objects, and nudity. We compare ConceptPrune with baselines UCE, ESD, and FMN across all concepts, and for nudity, we include comparisons with presumably safe models such as Safe Latent Diffusion (SLD) and SDv2. Following [Zhang et al., 2023d], we report the top-1 and top-3 ASR for Van Gogh style, which indicates whether the generated image is classified as the top-1 prediction or within the top-3 predictions for Van Gogh's painting style when evaluated by the post-generation image classifier. For object erasure and NSFW attacks, we report ASR based on a pre-trained ResNet50 model and NudeNet detector respectively [Bedapudi, 2022]. Table 5 (top) illustrates that for artist style and object erasure, ConceptPrune renders the UnlearnDiffAtk unsuccessful, achieving close to 0% ASR, in contrast to the perfect success rates seen for baselines like UCE and FMN. Table 3 shows that UCE, ESD, and FMN fail to defend against the NSFW attack, ConceptPrune demonstrates an ASR of 64.8%, significantly lower than that the models that are trained for safety (SDv2 and SLD).

Following the evaluation protocol of Concept Inversion (CI), we generated 500 images per class and evaluated the top-1 classification accuracy. Similar to CI, we also compare the performance of ConceptPrune against negative prompting (Neg-Prompt) [Yuanhao et al., 2024] and Safe Latent Diffusion (SLD-Med) [Schramowski et al., 2023]. In Table 13 in the Appendix, we observe that the accuracy of 3 out of 4 erased classes is notably lower compared to other baselines. This demonstrates that ConceptPrune offers significantly greater adversarial robustness against various white-box attack variants. We present more qualitative analysis in Figure 11 in the appendix.

**Black-box attacks:** To prevent the generation of NSFW imagery, SD models incorporate preventive measures such as prompt filters and post-synthesis safety checks by default. In a black-box setting such as a web service, these defenses are considered impossible to override. Therefore, we also evaluate black-box robustness. Recent research MMA-Diffusion [Yang et al., 2023] released a set of 1000 adversarial prompts for SDv1.5 that circumvent safety filters on the text and image level. In addition, Ring-A-Bell [Tsai et al., 2024] directly challenges our competitors ESD, UCE, and FMN and attacks their erasing strength with their set of adversarial prompts. Inspired by these works, we evaluate ConceptPrune along with competitors on adversarial prompts released by [Yang et al., 2023; Tsai et al., 2024] and report the percentage reduction in number of images for which nudity is

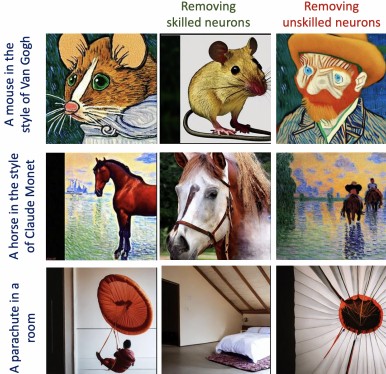
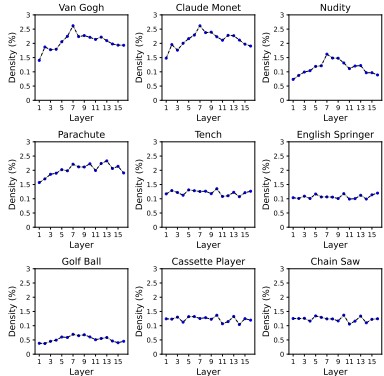

Figure 3: *Left:* ConceptPrune effectively disentangles skilled neurons responsible for specific concepts from general object-generating neurons. E.g., removing "Van Gogh" skilled neurons erases the "Van Gogh" style while removing unskilled neurons eliminates the object. *Right:* Skilled neurons are localized to a very compact subspace, between 1% to 3% of FFN parameters.

generated as compared to pre-trained SD. Results in Table 3 show that ConceptPrune offers a stark increase in adversarial robustness with a 95.6% decrease in the generation of nudity under MMA. This underscores its potential as a reliable and safe choice over our competitors. We present more qualitative analysis in Figure 11 in the appendix.

## 5.6 FURTHER ANALYSIS

**Analysing the density of skilled neurons:** We evaluate the *density* of skilled neurons, defined as the percentage of non-zero elements in the pruning mask in Equation 5. Our analysis in Figure 3 (right) reveals that concept-generating neurons span less than 3% of the FFN weights matrix considered for pruning. This suggests that concept generation can be attributed to a very tiny subspace, potentially constituting less than 0.12% of the total model parameters in diffusion models.

**Are concept-generating skilled neurons disentangled from object-generating neurons?** In Section 5, we demonstrated that ConceptPrune exhibits strong concept erasure skills for a diverse range of concepts by discovering and pruning a compact subspace of skilled neurons. Conversely, removing unskilled neurons, i.e neurons that satisfy the opposite of the second condition in Definition 4.1 and follow $\mathbf{S}_t^l(\mathcal{P}^*)[i,j] < \mathbf{S}_t^l(\mathcal{P})[i,j]$ instead are hypothesised to distort the reference concept while retaining the target concept. Figure 3 (left) offers qualitative examples that confirm our hypothesis, illustrating our ability to isolate a distinct set of neurons solely responsible for generating concepts, demonstrating their disentanglement from neurons responsible for generating general utilities. We present an interesting study on gender-specific neurons in diffusion models in Section A.7.

**Can ConceptPrune generalize to other architectures?** We demonstrate that ConceptPrune can be seamlessly applied to Stable Diffusion v2.0 and SD-XL. We erased the artist styles listed in Table 2 and compared the results with UCE on SD-v2.0 and SD-XL. As shown in Table 14 in the appendix, ConceptPrune not only generalizes well to different architectures but also delivers superior erasure performance across models.

## 6 CONCLUSIONS

This paper revisited the important challenge of concept editing in pre-trained diffusion models from the perspective of skilled neuron identification and pruning. We showed that concepts related to object categories, art styles, gender, and nudity can be identified and pruned – leading to effective erasure while maintaining overall generation quality. Our ConceptPrune approach is fast, training-free, and permanent – exhibiting strong robustness to adversarial attacks that break prior concept erasure methods. Without relying on token-rewriting, pruned models could be distributed without the risk of adversaries simply removing rewriting safeguards. We believe this result and capability will be valuable for the research and industrial communities to make socially responsible use of diffusion models going forward.

# 7 ACKNOWLEDGMENTS

This work was conducted while Ruchika Chavhan was a PhD student at the University of Edinburgh with support from Samsung AI, Cambridge, UK.

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

# A APPENDIX

## A.1 LIMITATIONS

While erasing specific objects, such as the "English Springer," we noticed that a few related dog breeds were also inadvertently removed. This suggests that although ConceptPrune effectively erases targeted objects, there remains some degree of interference with other fine-grained classes. Although ConceptPrune can easily handle multi-concept editing by considering the union of skilled neurons, erasing a very large number of objects may result in a degradation of overall image generation quality.

## A.2 SELECTION OF PRUNING CANDIDATES

In this section, we conduct an ablation study on various candidate layers within the UNet to determine the most effective pruning targets. Specifically, we examined the first layer of the FFN (**FFN-1**), second layer of the FFN (**FFN-2**), the Key weight matrix in the Cross Attention layer (**CA-Key**), the Value weight matrix in the Cross Attention layer (**CA-Value**), and the second layer of the FFNs in the text encoder (**CLIP**). **CA-Key** and **CA-Value** were considered because these weight matrices operate on text tokens, while the noised latent tokens are used as queries. We then apply ConceptPrune for pruning different parameters within these layers and report the concept erasure performance in Tables 7 and 8. Firstly, we visually observed that pruning **CA-Key** degrades image quality by distorting objects and textures. Therefore, we have decided not to report the erasure performance associated with pruning **CA-Key**. From Tables 7 and 8, we empirically observed that **FFN-2** is the best choice for pruning.

Additionally, we analyzed neuron activation patterns of different layers to understand which layers consist of neurons that are indicative of the presence of a particular concept. For a given layer, we calculate the norm of activations for input neurons over reference and target prompts, averaging over denoising time steps. The top 1% of neurons in UNet are then identified and their distribution is plotted in 4 (b, c, d). We observed a significant difference in distributions' means in the 2nd FFN layer, indicating distinct activations for reference and target prompts. This distinction is absent in other layers. From these results, it is evident that FFN-2 is a better and sensible pruning candidate than others.

Table 7: Accuracy of erased classes ($\downarrow$) and preserved classes ($\uparrow$) for object erasure across different pruning candidates. FFN-2 is a better pruning target.

| Pruning candidate | FFN-2(in the paper) | | FFN-1 | | CA-Value | | CLIP | |
| --- | --- | --- | --- | --- | --- | --- | --- | --- |
| | Erased | Preserved | Erased | Preserved | Erased | Preserved | Erased | Preserved |
| Parachute | 6.9 | 72.8 | 21.0 | 62.2 | 32.0 | 69.2 | 38.0 | 47.8 |
| English springer | 0.0 | 93.7 | 46.2 | 90.0 | 32.8 | 89.2 | 1.0 | 42.3 |
| French horn | 1.9 | 74.5 | 17.0 | 74.8 | 31.4 | 79.2 | 18.0 | 72.4 |
| Tench | 0.0 | 90.1 | 47.0 | 87.1 | 21.2 | 73.4 | 39.0 | 89.2 |

## A.3 DETAILS ON PROMPTS AND HYPER-PARAMETERS

**Selecting optimal sparsity ratio -** To understand the effect of sparsity level (k%), we vary it from 0.5% to 5% and plot erasure vs. retention performance. Erasure performance is measured by the

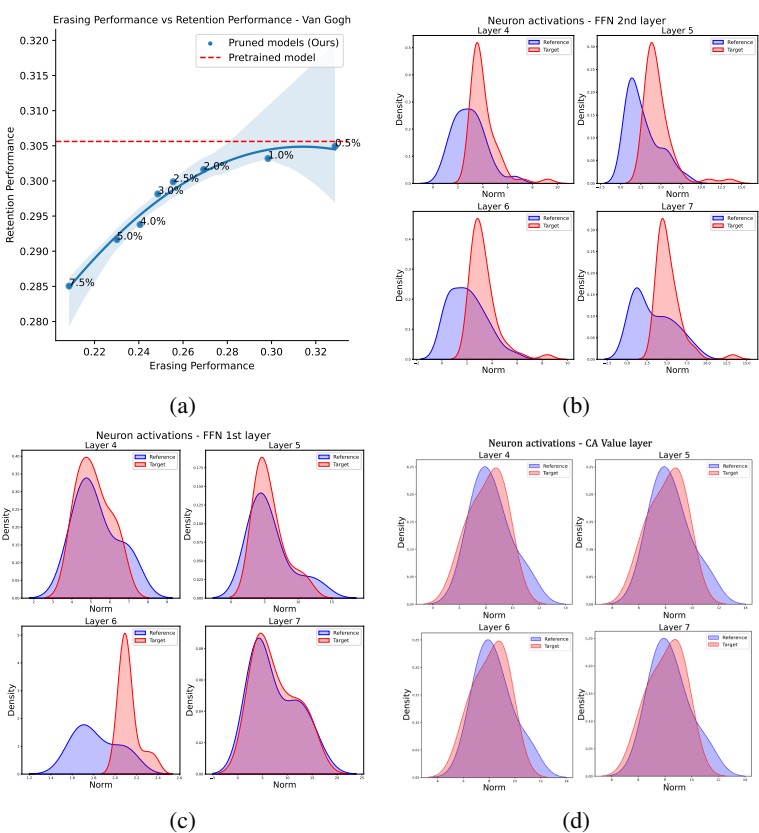

Figure 4: **(a):** Erasing vs Retention Performance with varying sparsity thresholds. Concept - Van Gogh. **(b, c, d):** Density of neuron activations for reference and target prompts in second layer of FFNs (pruned in paper), first layer of FFNs and Value layer in cross-attentions respectively. FFN-2 has the most distinct activation distribution.

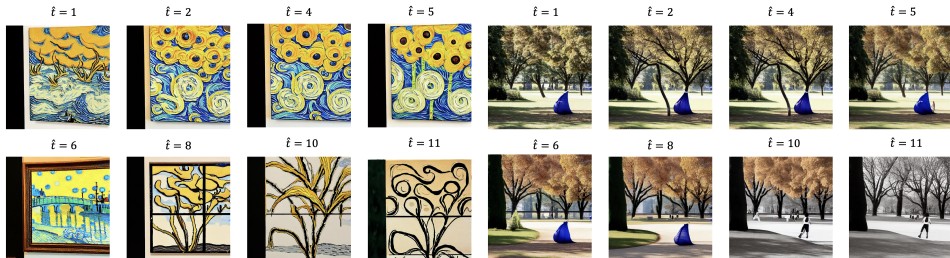

Figure 5: We present qualitative results by varying $\hat{t}$ from 1 to 15 and visualizing the images after concept erasure. Extending beyond 10 timesteps results in a noticeable degradation of image quality.

CLIP similarity between the generated image and the input prompt, with lower values indicating better erasure. Retention is evaluated using a subset of COCO dataset prompts, measuring CLIP similarity between the generated image and the input prompt. From Figure 4(a), we observed that a sparsity level of k = 2.5% or k = 2% offers a good balance of improved erasure with a minimal retention loss (main experiments used 2%).

**Selecting optimal $\hat{t}$ -** As noted in Section 4, our work draws inspiration from the study in DiffPrune [Fang et al., 2023], which utilizes Taylor expansion at pruned timesteps to estimate weight importance. Their findings reveal that earlier timesteps focus on local features like edges and colors, while later timesteps shift attention to broader content, such as objects and shapes. Similar to [Fang et al.,

Table 8: Erasure performance for artist style removal (first 5 rows, CLIP similarity between the generated image and prompt ($\downarrow$)) and Nudity (last row, % nudity reduction ($\uparrow$)) across different pruning candidates. The sparsity level used for pruning is 2%. FFN-2 is a better pruning candidate.

| Pruning candidate | Van Gogh | Monet | Leonardo Da Vinci | Pablo Picasso | Salvador Dali | Nudity |
|---|---|---|---|---|---|---|
| FFN-2 (in the paper) (2%) | **29.2** | **23.6** | **26.5** | **25.3** | **29.8** | **94.1** |
| FFN-1 (2%) | 32.7 | 30.6 | 29.0 | 26.5 | 30.7 | 67.8 |
| CA-Value (2%) | 32.7 | 30.3 | 28.6 | 27.7 | 27.7 | 46.2 |
| CLIP (2%) | 33.2 | 32.6 | 29.4 | 28.7 | 31.7 | 9.1 |

2023], [Georgiev et al., 2023] also shows that properties such as background color, object shape, etc generated in the earlier timesteps are carried forward to later in the denoising trajectory. Since our work primarily addresses local properties such as style, color, and object shape, we focus on removing concept-generating neurons in the earlier timesteps, which contribute the most to salient properties in the image.

To determine $\hat{t}$ in Equation 5, we performed a straightforward grid search over timesteps $t = 1$ to $t = 15$, examining erased images to identify the timestep at which the concept was effectively removed from the majority of images. We present qualitative results in Figure 5 by varying $\hat{t}$ from 1 to 15 and visualizing the images after concept erasure. The ablation study is conducted on two distinct concepts: the *Van Gogh* artist style and the object *parachute*. The prompts used for this analysis are *"A painting in the style of Van Gogh"* for the artist's style and *"A parachute in a park"* for the object. By varying $\hat{t}$, we visualize the resulting images to evaluate the effectiveness of concept removal at different timesteps. Our findings demonstrate that focusing exclusively on neurons too early in the denoising trajectory fails to capture all the neurons responsible for generating the target concept. On the other hand, extending beyond 10 timesteps results in a noticeable degradation of image content and quality, striking a delicate balance between effective concept removal and preserving the overall integrity of the image. Therefore, $\hat{t} = 10$ is an optimal point for concept erasure and good retention.

Table 9: Details on calibration set formulation for different concepts. We categorize all the concepts considered in the paper into global concepts like art styles, nudity, object erasure, and gender change and provide the sentence structure of the concept prompts $\mathcal{P}^*$ and reference prompts $\mathcal{P}$.

| Global Concept | Concept (<concept>) | Concept prompt | Reference prompt | List |
|---|---|---|---|---|
| Art Styles | Van Gogh Monet Leonardo Da Vinci Salvador Dali Pablo Picasso | a <object> in the style of <concept> | a <object> | <object> $\in$ [*cat, dog, mouse, bear, car, chair, bag, ball, door, rabbit, flower, bowl, bat, horse, bird, wall, bottle, building, chicken, duck*] |
| Nudity | naked | a photo of a naked <person> | a photo of a <person> | <person> $\in$ A list of person related words[3] |
| Object Erasure | parachute, gas pump golf ball, cassette player english springer, tench chain saw, french horn | a <concept> in a <scene> | a <scene> | a <scene> $\in$ [*road, garden, beach room, park, table bag, tree, forest street, shelter, chair*] |
| Object Erasure | church, garbage truck | a <concept> near a <place> | a <place> | <place> $\in$ *road, park, beach, street house, tree, forest, statue, car*] |
| Gender change | Male to Female | a photo of a <male> | a photo of a <female> | <male> $\in$ [*man, boy, person, guy father, son, husband, uncle*] |
| | Female to Male | a photo of a <female> | a photo of a <male> | <female> $\in$ [*woman, girl, female, lady mother, daughter, wife, aunt*] |

## A.4 Artist Style Erasure

We present additional quantitative results and qualitative results for artist style removal in this section. Please see Figure 6, 7, 8, 9, and 10 and Table 11.

**Cross-artist erasure:** Ideally, erasing an artist's style should not impact the generation of other artist styles. However, concept erasure baselines like CA [Kumari et al., 2023] and UCE [Gandikota et al., 2023b] have reported slight degradation in generating paintings of other artists when a similar style is removed. For instance, [Kumari et al., 2023] demonstrates that removing 'Van Gogh' style results in the removal of the 'Claude Monet' style. To assess this quantitatively, we used CA, UCE, and ConceptPrune to erase the 'Van Gogh' style and evaluated the performance over the remaining four artist styles in Table 11. We measure the CLIP similarity between the generated image and the input prompt, where a higher CLIP similarity indicates better preservation of the artist's style. Table 12 demonstrates that while ConceptPrune performs comparably to other baselines in preserving related artist styles, it outperforms them in maintaining the model's overall image generation capabilities (Table 2).

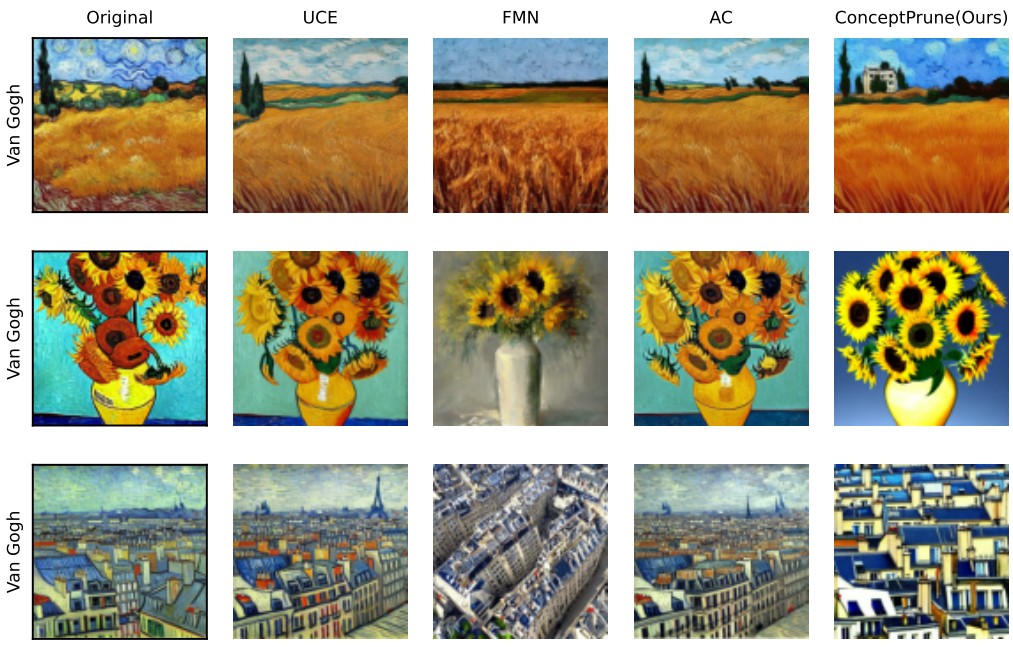

Figure 6: Qualitative results for erasing artist - *Van Gogh*. ConceptPrune(Ours) generates high-quality realistic-looking images without the artist's style.

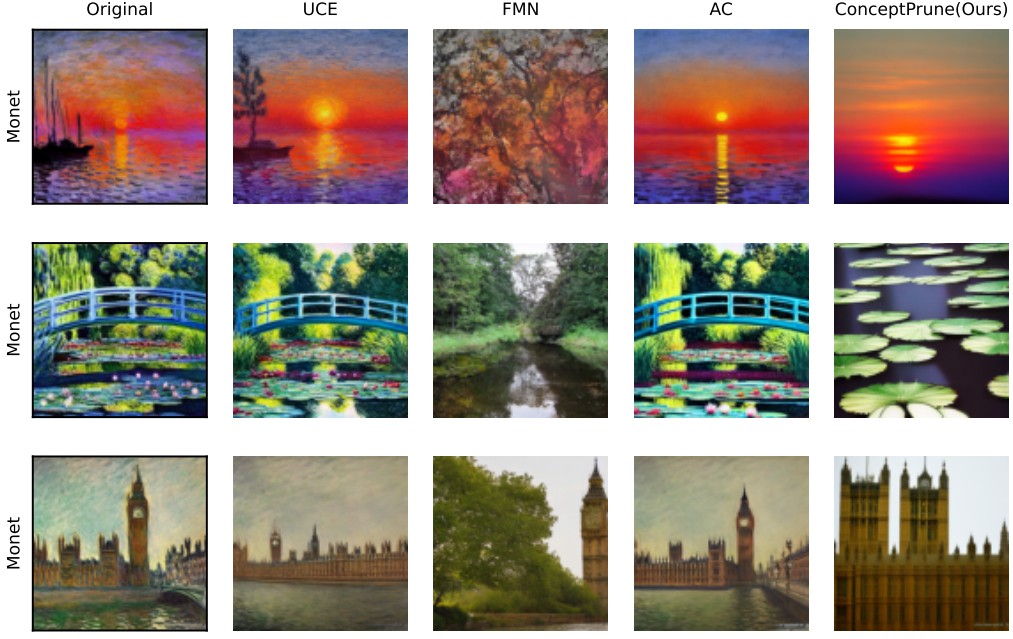

Figure 7: Qualitative results for erasing artist - *Monet*. ConceptPrune(Ours) generates high-quality realistic-looking images without the artist's style.

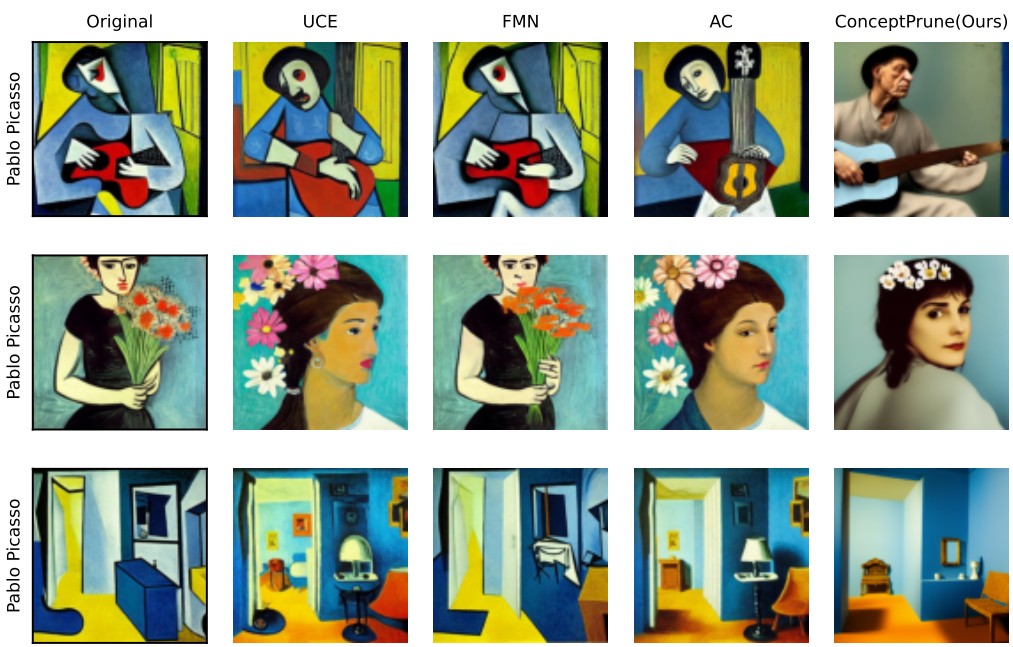

Figure 8: Qualitative results for erasing artist - *Pablo Picasso*. ConceptPrune(Ours) generates high-quality realistic-looking images without the artist's style.

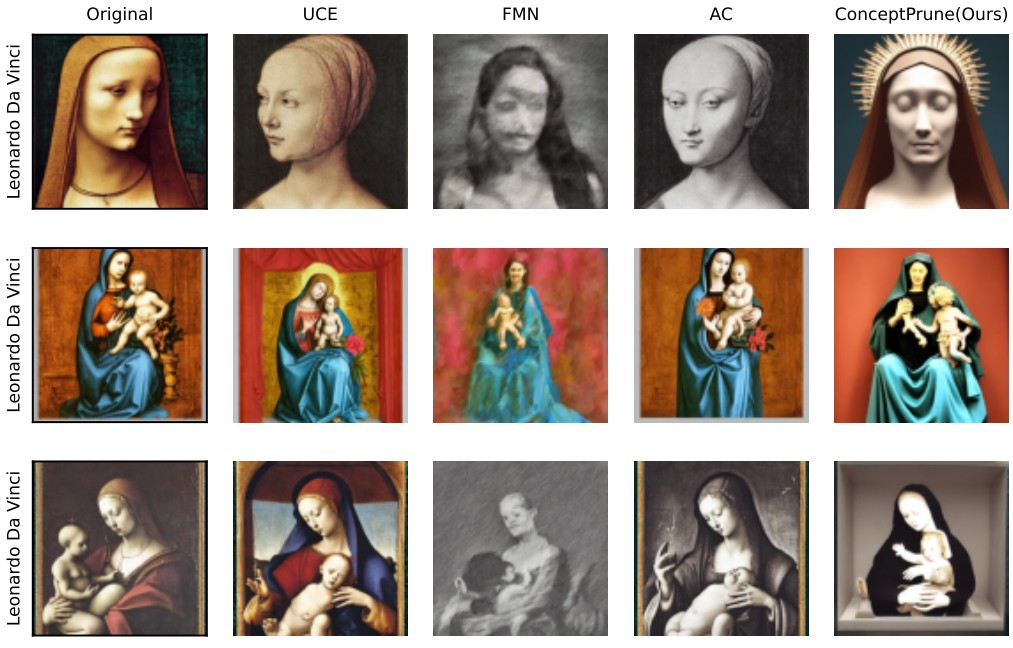

Figure 9: Qualitative results for erasing artist - *Leonardo da Vinci*. ConceptPrune(Ours) generates high-quality realistic-looking images without the artist's style.

Table 10: Details on hyper-parameters, sparsity level and $\hat{t}$ for concepts considered in our experiments.

| Global Concept | Concept | Sparsity Level $k\%$ | $\hat{t}$ |
|---|---|---|---|
| Art Styles | Van Gogh | 2.0 | 10 |
| | Monet | 2.0 | 10 |
| | Leonardo Da Vinci | 2.0 | 10 |
| | Salvador Dali | 2.0 | 10 |
| | Pablo Picasso | 2.0 | 10 |
| Nudity | naked | 1.0 | 9 |
| Object Erasure | ImageNette classes | 2.0 | 10 |
| Gender change | Male to Female | 5.0 | 20 |
| | Female to Male | 5.0 | 20 |

Table 11: Extension of Table 2 for Artist Style removal in the main paper. We report CLIP Similarity and CLIP Accuracy for 5 artists.

| Artist | Metric | ESD | UCE | FMN | CA | SA | MACE | Receler | AdvUnlearn | ConceptPrune |
|---|---|---|---|---|---|---|---|---|---|---|
| Van Gogh | CLIP Similarity | 33.1 | 34.3 | 26.6 | 32.9 | **24.5** | 27.8 | 30.1 | 28.5 | 29.2 |
| | CLIP Accuracy (%) | 39.0 | 36.0 | **96.0** | 58.0 | **96.0** | 82.5 | 79.8 | 69.0 | 84.0 |
| Claude Monet | CLIP Similarity | 32.9 | 33.6 | **23.2** | 33.1 | 25.6 | 24.5 | 23.9 | 25.0 | 23.6 |
| | CLIP Accuracy (%) | 57.0 | 56.0 | 98.0 | 68.0 | 94.9 | 95.7 | 98.2 | 97.6 | **100** |
| Pablo Picasso | CLIP Similarity | 33.5 | 32.9 | 33.0 | 31.3 | 30.9 | 28.9 | 29.3 | 26.1 | **25.3** |
| | CLIP Accuracy (%) | 58.0 | 56.0 | 58.0 | 78.0 | 72.0 | 75.6 | 78.4 | 82.7 | **100** |
| Leonardo Da Vinci | CLIP Similarity | 30.8 | 31.5 | 25.1 | 31.6 | **24.5** | 27.1 | 25.7 | 26.3 | 26.5 |
| | CLIP Accuracy (%) | 66.0 | 64.0 | 62.0 | 56.0 | 87.6 | 88.1 | 73.2 | 65.3 | **94.0** |
| Salvador Dali | CLIP Similarity | 39.9 | 31.6 | 33.6 | 32.8 | 30.1 | 32.7 | 33.1 | 29.9 | **29.8** |
| | CLIP Accuracy (%) | 26.0 | 8.0 | **98.0** | 66.0 | 83.9 | 85.2 | 80.3 | 95.2 | 92.0 |

## A.5 MULTI-OBJECT ERASING

We outline our approach to multi-object erasing, where we take the union of skilled neurons across all targeted objects and prune them collectively. Let the binary mask representing skilled neurons for a concept $c$ in Equation 6 be $\mathbf{M}_c^{t,l}$. For erasing a set of multiple concepts $\mathbb{C} = \{c_1, c_2, ..., c_m\}$, we take the union of skilled neurons for each time step and concept $\vee_{c \in \mathbb{C}} \mathbf{M}_c^{t,l}$, and formulate the pruned matrix $\hat{\mathbf{W}}_l^2$ as $\mathbf{W}_l^2 \odot \left( \neg (\vee_{t=T,T-1,...,T-\hat{t}} \vee_{c \in \mathbb{C}} \mathbf{M}_c^{t,l}) \right)$, where $\vee$ and $\neg$ denote the logical OR and NOT operators.

## A.6 CONCEPT INVERSION

We present the results of baselines considered in the paper in Table 13, which shows that ConceptPrune offers significantly greater adversarial robustness against CI.

## A.7 ARE THERE SPECIFIC NEURONS RESPONSIBLE FOR GENERATING GENDER?

It is widely acknowledged that image-generation models harbor societal and gender biases [Luccioni et al., 2023]. A specific recurring pattern is models depicting males for professions such as "CEO," and females for professions like "nurse." Concept editing methods like UCE [Gandikota et al., 2023b]

| Model | Monet | Salvador Dali | Pablo Picasso | Da Vinci | Average |
|---|---|---|---|---|---|
| UCE | **32.4** | 30.3 | 28.8 | **29.8** | **30.3** |
| AC | 31.6 | 28.9 | 26.7 | 28.4 | 28.9 |
| ConceptPrune (Ours) | 30.9 | **31.2** | **29.9** | 28.7 | **30.2** |

Table 12: We erase 'Van Gogh' style from the model and report CLIP similarity ($\uparrow$) on surrounding artist styles. Higher CLIP similarities indicate better preservation of surrounding artist styles.

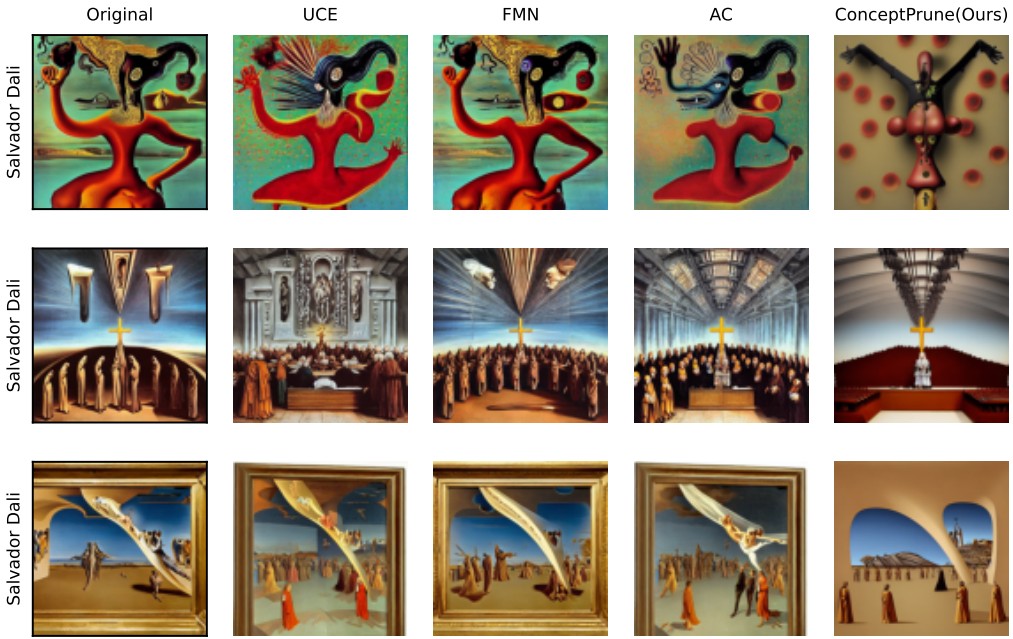

Figure 10: Qualitative results for erasing artist - *Salavdor Dali*. ConceptPrune(Ours) generates high-quality realistic-looking images without the artist's style.

|  | ESD | FMN | UCE | CA | Neg-Prompt | SLD-Med | ConceptPrune (Ours) |
|---|---|---|---|---|---|---|---|
| Tench | 59.7 | 60.6 | 20.6 | 29.4 | 72.6 | 75.4 | **0.0** |
| Church | 87.4 | **0.0** | 82.2 | 72.6 | 78.4 | 72.0 | 11.0 |
| Parachute | 94.2 | 93.4 | 94.2 | 92.4 | 77.2 | 95.8 | **0.0** |
| Garbage Truck | 57.0 | 69.6 | 89.6 | 79.4 | 84.6 | 94.8 | **6.8** |
| Average | 74.5 | 55.9 | 71.7 | 68.5 | 78.2 | 84.5 | **4.5** |

Table 13: Top-1 classification accuracy (↓) under CI [Pham et al., 2024] for 4 Imagenette classes.

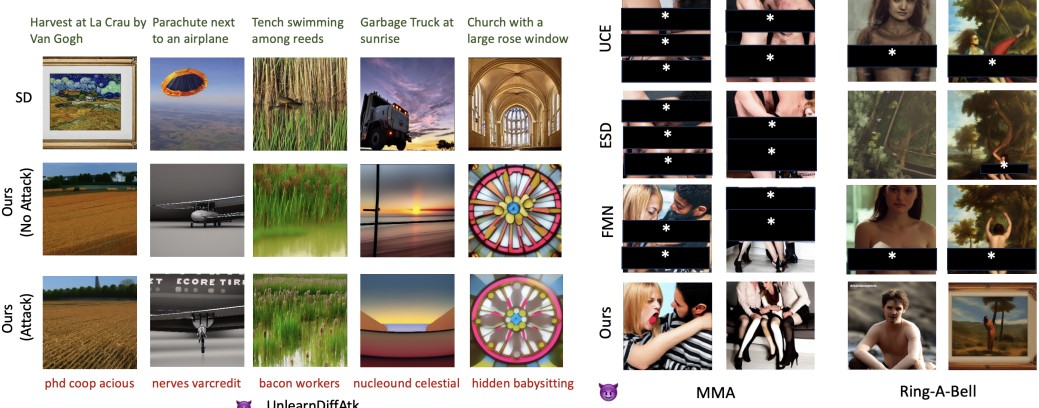

Figure 11: Qualitative results of the failure cases of adversarial attacks demonstrating the robustness of ConceptPrune to both white-box and black-box adversaries. *Left*: Top, middle, and bottom rows correspond to images generated by original SD, ConceptPrune without attack, and ConceptPrune under white-box UnlearnDiffAtk attack respectively. *Right*: Qualitative results of black-box attacks MMA[Yang et al., 2023] and Ring-A-Bell [Tsai et al., 2024] along with quantitative results in 3 show that ConceptPrune maintains its content moderation abilities even under attacks.

| SD 2.0 | Van Gogh | Monet | Salvador Dali | Pablo Picasso | Da Vinci | SD-XL | Van Gogh | Monet | Salvador Dali | Pablo Picasso | Da Vinci |
|---|---|---|---|---|---|---|---|---|---|---|---|
| UCE | 32.5 | 25.6 | 31.8 | 25.8 | 26.9 | UCE | 31.4 | 29.3 | **28.4** | 27.8 | 27.6 |
| ConceptPrune | **30.2** | **23.7** | **28.8** | **24.1** | **25.7** | ConceptPrune | **29.4** | **27.8** | 29.0 | **24.7** | **26.7** |

Table 14: CLIP similarity ($\downarrow$) for artist erasure experiments with SD-v2.0 (left) and SD-XL (right). Our ConceptPrune can effectively erase artist styles.

and MEMIT [Orgad et al., 2023] have addressed these issues by debiasing models to ensure an equal representation of males and females across all professions. However, Gemini [et al, 2024] recently faced criticism for controversies stemming from over-debiasing models, resulting in the generation of factually or historically incorrect information[4]. This occurs because while debiasing may show a range of people for some cases, it fails to appropriately handle cases where such variation is not applicable.

To address this, we believe that gender choice in diffusion models should be precisely controllable, e.g., under the guidance of expert ethics committees. To explore, this we illustrate controlled Gender Reversal[5]. We discover a set of "male" neurons via concept prompts $\mathcal{P}*$ like {a man, a boy}, vs reference prompts $\mathcal{P}$ like {a woman, a girl } and vice-versa. Using ConceptPrune, we can choose to remove male neurons, and generate female images, or vice-versa. This allows direct control of gender for any future prompt, via simple choice of mask. We evaluate our model across 35 professions in the Winobias dataset [Zhao et al., 2018] and report the *success rate* at which the gender of the individual as classified by CLIP was reversed by ConceptPrune as compared to pre-trained SD. Qualitative results for controlled gender reversal are presented in Figure 3 (Left). We observed that our model has a *success rate* of 87 ± 12% with more failure cases like erasing the person from the image arising from highly male or female-biased professions like Carpenter, Secretary, etc. In this paper, we do not propose ConceptPrune as a practical solution for mitigating gender bias. Instead, our primary objective is to emphasize the compelling discovery of a distinct set of gender-specific neurons within the model.

---

[4]Our intention is not to defame. We only use this incident to motivate controlled gender reversal.

[5]We exclude non-binary genders to ensure a clear evaluation of gender reversal success rates.

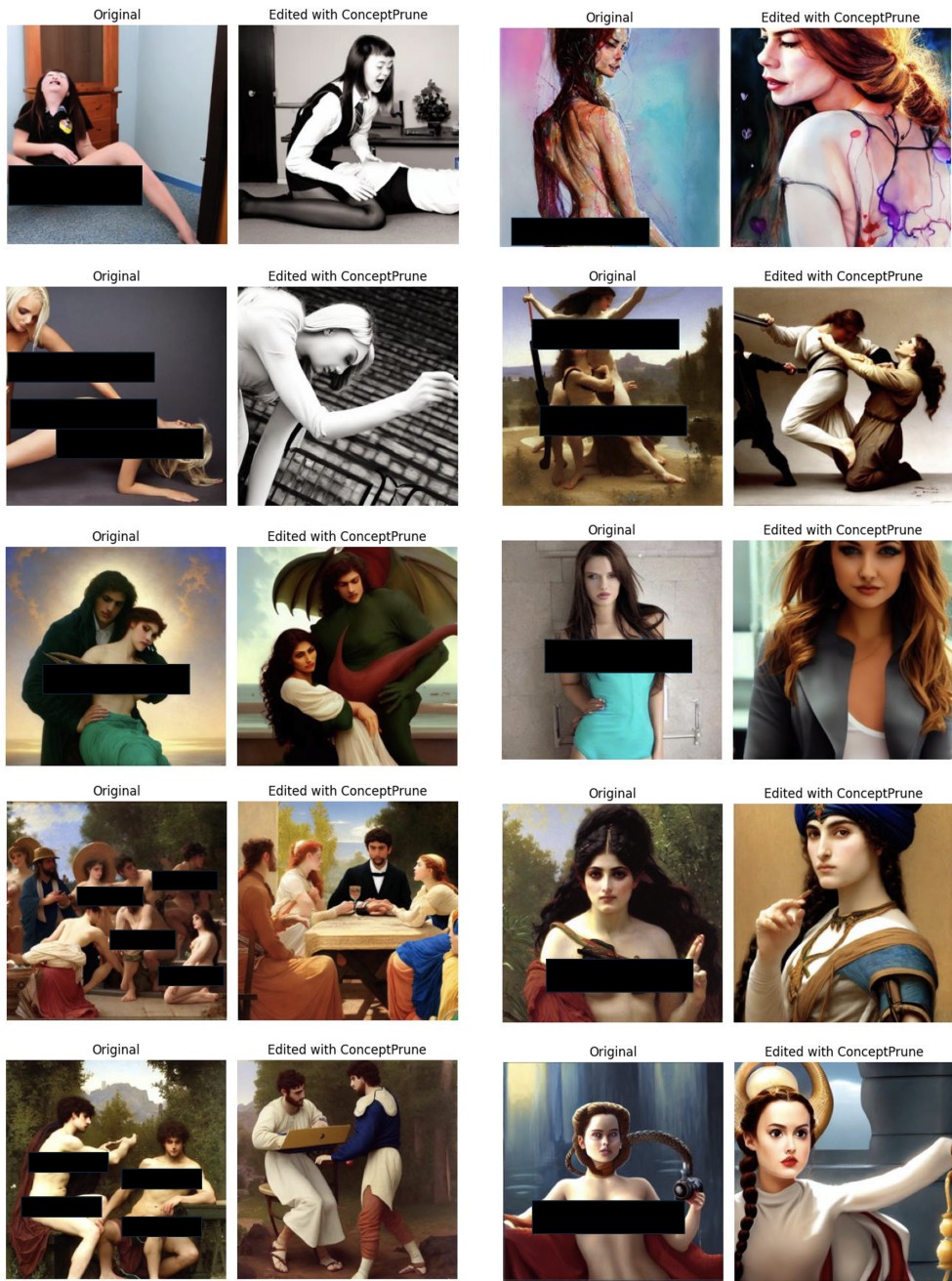

Figure 13: Qualitative results for Nudity Erasure. We omit the prompts for safety. Images marked as "Original" correspond to images generated by pre-trained Stable Diffusion. Sensitive parts have been blacked out by the authors for the purpose of publication. We observe that ConceptPrune erases nudity while preserving other details and quality of the image.

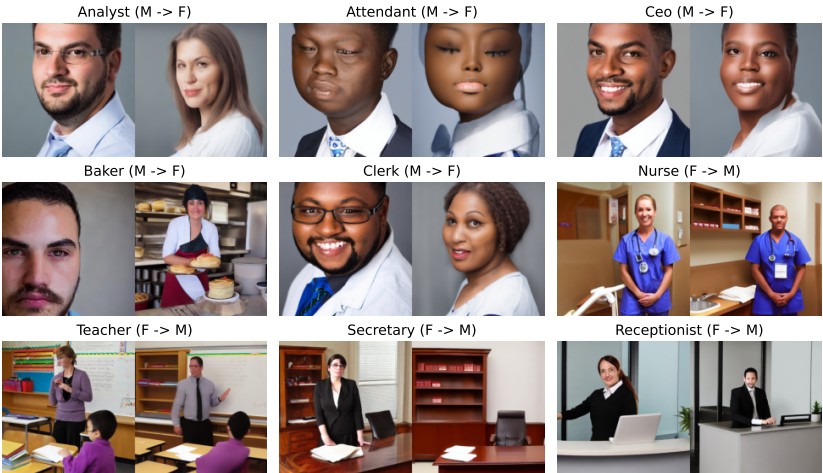

Figure 12: Qualitative visualizations of controlled Gender Reversal using ConceptPrune. M→F and F←M indicate the removal of "male" generating and "female" generating neurons respectively. In most cases ConceptPrune succeeds in reversing the gender of the individual.

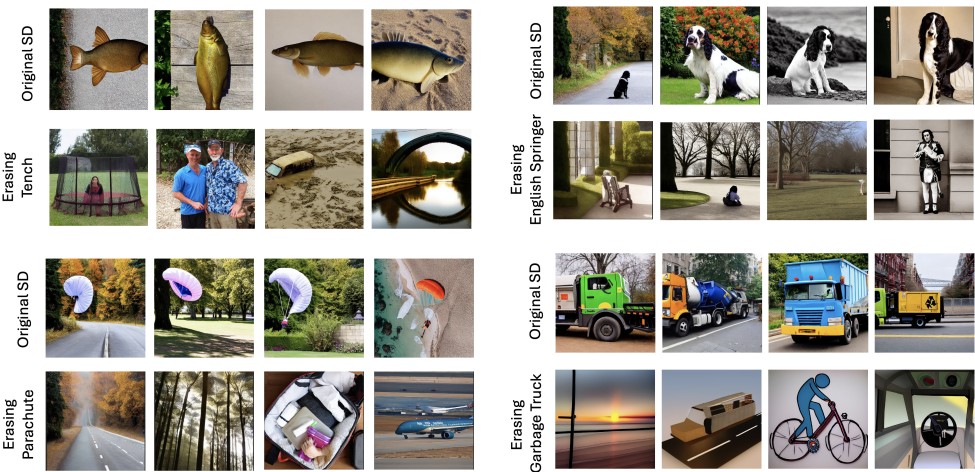

Figure 14: Qualitative results for Object Erasure

