# OpenReview forum: "ConceptPrune: Concept Editing in Diffusion Models via Skilled Neuron Pruning"
_ICLR.cc/2025/Conference — ICLR 2025 Poster_

### Official Review · Reviewer_uDZE · 2024-11-03

**Soundness:** 3
**Presentation:** 2
**Contribution:** 2
**Rating:** 3
**Confidence:** 5

**Summary:**

This paper addresses the risks of large-scale text-to-image diffusion models, including misuse for unsafe content, copyright violations, and bias perpetuation. Traditional methods for removing undesired concepts from these models often require intensive fine-tuning or token remapping, which remain vulnerable to adversarial jailbreaks. To tackle these issues, the authors propose ConceptPrune, a training-free approach that locates key regions in pre-trained models linked to unwanted concepts and applies weight pruning for efficient concept unlearning. Experiments show ConceptPrune effectively erases specific styles, nudity, and objects by pruning only 0.12% of weights, allowing multi-concept removal with resilience to adversarial attacks.

**Strengths:**

1. The paper is clearly written and well-structured.
2. It offers an innovative approach to unlearning by employing model pruning techniques.

**Weaknesses:**

1. There is a lack of adequate baselines. The authors only compare their method to a few early baselines in machine unlearning within diffusion models. As far as I know, many more effective methods have emerged recently, but these were not included in the comparison.

2. The robustness of the method is insufficiently studied. According to recent research, machine unlearning can be vulnerable to adversarial prompts. Without a robust evaluation, it remains unclear whether pruning-based unlearning is indeed resilient against such attacks.

**Questions:**

1. Could the authors add more baselines by comparing their method with recently published machine unlearning methods in diffusion models?

2. Could the authors include experiments on the robustness of machine unlearning? Specifically, experiments involving adversarial prompts or similar robustness tests would be valuable.

---

> ### Author Response · Authors · 2024-11-27
> **Response to Reviewer uDZE**
>
> Thank you for your review. We answer your concerns below.
>
> - **Lack of adequate baselines**
>
> Please refer to the common response. While we have tried our best to include as many related works as possible in the revised version of the paper, it is unclear while baselines the revieweer has implied. We would appreciate a more specific review.
>
>
> **Robustness of the method is insufficiently studied.**
>
> We acknowledge the reviewer’s observation that recent studies, such as [1,2], have highlighted the vulnerability of machine unlearning methods to adversarial prompts. However, we respectfully disagree with the claim that the robustness of our method has not been sufficiently studied. We have conducted an extensive evaluation of **ConceptPrune** against two black-box and two white-box adversarial attacks across tasks involving artist, nudity, and object erasure. Our results clearly demonstrate that **ConceptPrune** exhibits superior adversarial robustness compared to its direct competitors.
>
> [1] Jailbreak Attacks and Defenses against Multimodal Generative Models: A Survey, Liu et al, arXiv
> [2] Adversarial attacks and defenses on text-to-image diffusion models: A survey, Zhang et al, arXiv

---

### Official Review · Reviewer_YHCm · 2024-11-04

**Soundness:** 3
**Presentation:** 3
**Contribution:** 2
**Rating:** 6
**Confidence:** 4

**Summary:**

The paper introduces ConceptPrune, a method for editing concepts in pre-trained diffusion models by pruning neurons associated with undesired concepts, like specific artistic styles, nudity, and biases. This approach identifies "skilled neurons" within diffusion models, such as Stable Diffusion, that activate in response to specific target concepts. By selectively pruning these neurons, the method effectively removes the unwanted concepts without the need for extensive fine-tuning, maintaining the model's overall performance on unrelated tasks. ConceptPrune demonstrates significant resistance to both black-box and white-box adversarial attacks, showing promise as a scalable and robust approach to safer model deployment.

**Strengths:**

Strengths:
- The investigation into how specific weights influence certain abilities in text-to-image (T2I) models is intriguing and provides valuable insights into the inner workings of these models.
- The paper presents extensive experiments demonstrating the effectiveness of the proposed method across various editing targets, including nudity, artistic style, and specific objects.
- Additionally, the paper is well-structured with a clear logical flow, making it easy to follow and understand.

**Weaknesses:**

- **Limited Comparison with Recent SOTA Methods**: The paper lacks sufficient discussion and comparison with recent state-of-the-art methods. For instance, [1] appears to be a strong baseline, demonstrating both high robustness in concept erasure and good utility preservation. While it’s acceptable if this paper does not outperform [1], it would be beneficial to include [1] as a baseline and discuss potential improvements for the current approach in light of their findings.

- **Insufficient Discussion on Related Concept Pruning Work**: The paper lacks sufficient background on concept pruning and similar work. Several studies ([2, 3, 4]) have explored the relationship between weight importance and model abilities during unlearning and editing tasks. Incorporating these works would help situate this paper within the broader landscape and highlight how ConceptPrune builds on or differs from these existing approaches.

[1] Zhang, Yimeng, et al. "Defensive Unlearning with Adversarial Training for Robust Concept Erasure in Diffusion Models." arXiv preprint arXiv:2405.15234 (2024).
[2] Fan, Chongyu, et al. "Salun: Empowering machine unlearning via gradient-based weight saliency in both image classification and generation." arXiv preprint arXiv:2310.12508 (2023).
[3] Wu, Jing, and Mehrtash Harandi. "Scissorhands: Scrub Data Influence via Connection Sensitivity in Networks." arXiv preprint arXiv:2401.06187 (2024).
[4] Foster, Jack, Stefan Schoepf, and Alexandra Brintrup. "Fast machine unlearning without retraining through selective synaptic dampening." Proceedings of the AAAI Conference on Artificial Intelligence. Vol. 38. No. 11. 2024.

**Questions:**

1. Could you please add discussions on the recent SOTA methods and relevant baselines as highlighted in the weaknesses section?

2. How did you determine the optimal pruning ratio for skilled neurons? Please describe the considerations or criteria used in selecting a suitable pruning ratio.

---

> ### Author Response · Authors · 2024-11-27
> **Response to reviewer YHCm**
>
> We thank the reviewer for their constructive feedback and appreciate the effort put into the review. We address the reviewer's concerns below.
>
>
> - **Limited Comparison with Recent SOTA Methods and Insufficient Discussion on Related Concept Pruning Work**
>
> We appreciate the reviewer for bringing these two recent works to our attention. Given the numerous recent state-of-the-art (SOTA) methods under consideration, we propose a categorization of related work based on their applicability to real-world scenarios.
>
> We categorized all baselines based on whether they are training-free or lightweight, as these two are important criteria for a real-world application. We direct the reviewer to the common response and Section 2 of the paper for further details. Based on this, the requested baselines - Scissorhands, SalUn, and AdvUnlearn can be considered indirect competitors, as ConceptPrune enables efficient erasing concepts without the need for fine-tuning or backpropagation, unlike these two methods.
>
> We have updated Table 2 (artist erasure), Figure 2 and Table 3 (nudity erasure), and Table 5 (adversarial attacks) to incorporate the new baselines. We observe that while ConceptPrune outperforms SalUn in adversarial robustness (Table 3 and 5), Scissorhands and AdvUnlearn are more adversarially robust than the two. However, ConceptPrune is more adversarially robust than its direct competitors.
>
>
> - **Optimal Pruning ratio for skilled neurons**
>
> We direct the reviewer to Section A.2 in the appendix, where we understand the effect of the pruning ratio on the erasure vs retention trade-off. For our experiments, we pick the pruning ratio that offers a good balance of improved erasure with minimal retention loss.
>
>
> **We have addressed all the comments and concerns raised by the reviewer. We sincerely hope that you consider revising the score accordingly.**

---

### Official Review · Reviewer_PDbh · 2024-11-05

**Soundness:** 3
**Presentation:** 3
**Contribution:** 3
**Rating:** 8
**Confidence:** 5

**Summary:**

To prevent the misuse of text-to-image diffusion models, this paper proposes a training-free approach using weight pruning for unlearning undesired concepts, such as artistic styles, nudity, and specific objects.

**Strengths:**

1. The experiments cover all diffusion model unlearning tasks, including artistic styles, nudity, and specific objects.
2. Consider ASR against different attacks while simultaneously evaluating FID and CLIP score to assess model utility.
3. The proposed method requires no additional training.

**Weaknesses:**

1. The chosen diffusion model baselines are weak and not state-of-the-art methods, so the comparison does not effectively demonstrate true superiority. To defend against adversarial prompt attacks, a stronger baseline, such as AdvUnlearn [1], should be considered.
2. More visualization examples are needed, as the current version only includes visualizations for the style unlearning task.

[1]  Defensive Unlearning with Adversarial Training for Robust Concept Erasure in Diffusion Models, NeurIPS’24

**Questions:**

In Table 5, the FID metric is missing, and the performance of the base model (SD) should be included for greater clarity.

---

> ### Author Response · Authors · 2024-11-27
> **Response to Reviewer PDbh**
>
> We thank the reviewer for their constructive feedback and appreciate the effort put into the review. We address the reviewer's concern below.
>
> - **Comparison with AdvUnlearn**
>
> We appreciate the reviewer for bringing recent works to our attention. Given the numerous recent state-of-the-art (SOTA) methods under consideration, we propose a categorization of related work based on their applicability to real-world scenarios.
>
> We categorized all baselines based on whether they are training-free or lightweight, as these two are important criteria for a real-world application. We direct the reviewer to the common response and Section 2 of the paper for further details. We point out that AdvUnlearn relies on extensive fine-tuning with adversarial prompts, whereas ConceptPrune eliminates concepts without requiring any fine-tuning or backpropagation. From Table 5, although AdvUnlearn is more robust than other baselines, ConceptPrune is more adversarially robust to its direct competitors.
>
> - **Visualizations**
>
> We also direct the reader to the appendix where we have provided many visualization examples in the appendix on artist erasure (Figures 5, 6, 7, 8, 9), nudity erasure (Figure 12), and object erasure (Figure 13). We also present qualitative results for white-box and black-box attacks (Figure 10).
>
> - **In Table 5, the FID metric is missing**
>
> Table 5 (now Table 6) on multi-concept erasure already contains the FID on 3k samples in the COCO dataset to demonstrate that we are comparable to UCE in retaining unrelated concepts.
>
>
> **We have addressed all the comments and concerns raised by the reviewer. We sincerely hope that you consider revising the score accordingly. Thank you for your time.**

---

> > ### Comment · Reviewer_PDbh · 2024-12-01
> >
> > The concerns have been addressed, and I will be increasing the rating. Thank you to the authors for their response and the revised submission.

---

### Official Review · Reviewer_B4VY · 2024-11-06

**Soundness:** 2
**Presentation:** 3
**Contribution:** 2
**Rating:** 6
**Confidence:** 5

**Summary:**

The paper introduces ConceptPrune, a training-free method for concept editing in pre-trained diffusion models, specifically latent diffusion models like Stable Diffusion. The core idea is to identify and prune "skilled neurons" within the feed-forward networks (FFNs) of the model's UNet architecture that are responsible for generating undesirable concepts. By calculating importance scores based on neuron activations for target (undesired) and reference (desired) prompts, the method isolates neurons that predominantly influence the generation of unwanted content. Pruning these neurons effectively erases the target concepts while preserving the model's ability to generate unrelated content. The authors demonstrate the efficacy of ConceptPrune across various concepts, including artistic styles, nudity, object erasure, and gender biases, showing that pruning approximately 0.12% of the model's weights suffices. Additionally, the method exhibits robustness against both white-box and black-box adversarial attacks aimed at circumventing concept removal.

**Strengths:**

1. The paper introduces an application of neuron pruning techniques to concept editing in diffusion models. Specifically, ConceptPrune utilizes the identification and pruning of skilled neurons, providing an alternative approach to mitigating undesired content generation.

2. The experimental results support the method's effectiveness in erasing various concepts while maintaining image generation quality. The authors compare ConceptPrune with several baselines, demonstrating its performance in concept removal and robustness against adversarial attacks.

3. The manuscript is well-written and well-structured, offering clear explanations of the methodology and detailed descriptions of the experiments. Mathematical formulations are provided to illustrate the approach, and visual examples enhance comprehension.

4. The work addresses significant ethical concerns related to the generation of unsafe or copyright-infringing content in diffusion models.

**Weaknesses:**

1. The method seems relatively straightforward and may be considered incremental in light of existing works [1,2] that also employ pruning techniques for unlearning in diffusion models. It would strengthen the contribution of the paper to clearly delineate the differences between the proposed approach and these related methods. Specifically, methods such as [1,3] have achieved state-of-the-art results in diffusion model unlearning tasks. Including these methods as baselines in the experimental comparisons would provide a more comprehensive evaluation and help illustrate the effectiveness of the proposed scheme.

2. The experimental validation is conducted solely on Stable Diffusion, which may limit the generalizability of the proposed method. To address this potential limitation, it is recommended that the effectiveness of the approach be demonstrated on additional diffusion models, such as [4,5]. This would provide evidence that the method is not tightly coupled to a specific architecture and can be broadly applied to other models in the domain.

3. The choice of focusing on the second layer of the feed-forward networks (FFN-2) for pruning appears to be based on empirical observations and may depend significantly on the architecture of Stable Diffusion. Is there theoretical justification or empirical evidence supporting this selection as the optimal pruning target? Providing analytical insights or additional experiments on different architectures would help ascertain whether this observation holds true universally or is specific to the model under consideration.

4. The decision to aggregate skilled neurons over the last 10 timesteps for pruning seems somewhat arbitrary. Could the authors elaborate on the reasoning behind selecting this particular range of timesteps? It may be beneficial to reference insights from related literature [6,7] that discuss the importance of different timesteps in the denoising process. A more thorough explanation or an exploration of how varying the number of timesteps affects the results would enhance the understanding of this methodological choice.

5. The evaluation of unlearning effectiveness in the context of artistic style erasure relies on CLIP-based similarity metrics. However, CLIP-based metrics consider redundant factors, not only the style but also the object content, arrangement, etc. It is potentially confounding the assessment of style erasure alone. This approach may not effectively isolate the stylistic elements from other irrelevant factors. It is recommended to employ a dedicated style classifier for evaluation (e.g., classifier in [8]) to more accurately measure the degree of style removal. This would provide a more reliable and focused assessment of the method's effectiveness in erasing artistic styles.

> [1] Wu, Jing, and Mehrtash Harandi. "Scissorhands: Scrub Data Influence via Connection Sensitivity in Networks." arXiv preprint arXiv:2401.06187 (2024).
>
> [2] Yang, Tianyun, Juan Cao, and Chang Xu. "Pruning for Robust Concept Erasing in Diffusion Models." arXiv preprint arXiv:2405.16534 (2024).
>
> [3] Fan, Chongyu, et al. "Salun: Empowering machine unlearning via gradient-based weight saliency in both image classification and generation." arXiv preprint arXiv:2310.12508 (2023).
>
> [4] Podell, Dustin, et al. "Sdxl: Improving latent diffusion models for high-resolution image synthesis." arXiv preprint arXiv:2307.01952 (2023).
>
> [5] Saharia, Chitwan, et al. "Photorealistic text-to-image diffusion models with deep language understanding." Advances in neural information processing systems 35 (2022): 36479-36494.
>
> [6] Balaji, Yogesh, et al. "ediff-i: Text-to-image diffusion models with an ensemble of expert denoisers." arXiv preprint arXiv:2211.01324 (2022).
>
> [7] Georgiev, Kristian, et al. "The journey, not the destination: How data guides diffusion models." arXiv preprint arXiv:2312.06205 (2023).
>
> [8] Zhang, Yihua, et al. "Unlearncanvas: A stylized image dataset to benchmark machine unlearning for diffusion models." arXiv preprint arXiv:2402.11846 (2024).

**Questions:**

See weaknesses.

---

> ### Author Response · Authors · 2024-11-27
> **Response to Reviewer B4VY**
>
> We thank the reviewer for their constructive feedback and appreciate the effort put into the review. We respond to all comments below.
>
> - **The method seems relatively straightforward and may be considered incremental in light of existing works [1,2] that also employ pruning techniques for unlearning in diffusion models. Comparison to more SOTA diffusion model unlearning tasks**
>
> We appreciate the reviewer for bringing these two recent works to our attention. Given the numerous recent state-of-the-art (SOTA) methods under consideration, we propose a categorization of related work based on their applicability to real-world scenarios.
>
> We categorized all baselines based on whether they are training-free or lightweight, as these two are important criteria for a real-world application. We direct the reviewer to the common response and Section 2 of the paper for further details. Based on this, Scissorhands and SalUn can be considered indirect competitors, as ConceptPrune enables efficient erasing concepts without the need for fine-tuning or backpropagation, unlike these two methods.
>
> We have updated Table 2 (artist erasure), Figure 2 and Table 3 (nudity erasure), and Table 5 (adversarial attacks) to incorporate the new baselines. We observe that while ConceptPrune outperforms SalUn in adversarial robustness (Table 3 and 5), Scissorhands is more adversarially robust than the two. However, ConceptPrune is more adversarially robust than its direct competitors.
>
> We agree that our work is related  to [2] as they also employ a pruning method, but we could not perform any comparisons die to lack of open-source code or models.
>
> - **Choice of FFN-2**
>
> Our work draws inspiration from several papers that have effectively isolated skilled neurons within FFNs [A-C], particularly [B], which demonstrated that FFNs in LLMs implicitly encode mixture models, allowing specific skills or concepts to be activated as components of these mixtures. While we acknowledge the existence of other studies focusing on concept control via attention layers, a potential reason for the observed difference is that skills may be more disentangled in FFN layers [B]. Consequently, while it is possible to densely modulate attention layers, selective pruning for concept control is most effective in FFN layers, where the disentangled structure makes it easier to isolate and prune specific concepts.
>
> [A] Finding Skill Neurons in Pre-trained Transformer-based Language Models, Wang et al, EMNLP 2022
> [B] Emergent Modularity in Pre-trained Transformers, Zhang et al, ACL 2023
> [C] Knowledge Neurons in Pretrained Transformers, Dai et al, ACL 2022
>
> In our study, we identify pruning candidates for SD1.4/1.5 through a comprehensive ablation analysis. This analysis evaluates various potential candidates, including FFN-2 (as used in the paper), FFN-1 (the first layer of the FFN), the Cross-Attention Value Matrix (CA-Value), and both text encoders within the pipeline (CLIP). To validate our findings, we repeat this ablation study on SD-XL, confirming that FFN-2 consistently emerges as the optimal pruning candidate for effective concept erasure.
>
> Artist Styles - We report CLIP similarity between the generated image and input prompt for different pruning candidates for 5 artist styles. Lower CLIP similarity indicates better concept erasure.
>
> | **Pruning Candidate**           | **FFN 2nd layer (in the paper) (2%)** | **FFN-1 (2%)** | **CA-Value (2%)** | **CLIP (original text encoder) (2%)** | **Second text encoder (2%)** |
> |----------------------------------|---------------------------------------|----------------|-------------------|---------------------------------------|-----------------------------|
> | Van Gogh                        |**29.4**                                  | 31.4           | 31.4              | 32.3                                  | 31.7                        |
> | Monet                           | **27.8**                                 | 30.2           | 30.6              | 33.1                                  | 34.5                        |
> | Leonardo Da Vinci               | **26.7**                                  | 29.7           | 28.7              | 27.2                                  | 29.4                        |
> | Pablo Picasso                   | **24.7**                                  | 27.8           | 26.5              | 27.1                                  | 29.8                        |
> | Salvador Dali                   | **29.0**                                | 31.1           | 30.3              | 31.3                                  | 31.6                        |

---

> ### Author Response · Authors · 2024-11-27
> **Author response continued**
>
> - **Generalization to other architectures**
>
> To demonstrate Concept-Prune’s generalization to other architectures, we add more results on SD-XL in Table 13 in the Appendix. We erased different artist styles with UCE for comparison and demonstrated that for both SD 2.0 and SD-XL, ConceptPrune not only generalizes to different architectures but also delivers superior erasure performance.
>
> - **Aggregation over timesteps**
>
> Thank you for the insightful question, and we sincerely appreciate the reviewer highlighting these interesting papers. As noted in Section 4, our work draws inspiration from the study in DiffPrune [9], which utilizes Taylor expansion at pruned timesteps to estimate weight importance. Their findings reveal that earlier timesteps focus on local features like edges and colors, while later timesteps shift attention to broader content, such as objects and shapes. Similar to [9], [10] also shows that properties as background color, object shape, etc generated in the earlier timesteps are carried forward to later in the denoising trajectory. Since our work primarily addresses local properties such as style, color, and object shape, we focus on removing concept-generating neurons in the earlier timesteps, which contribute the most to salient properties in the image.
>
> To determine $\hat{t}$ in Equation 6, we performed a straightforward grid search over timesteps t=1 to t=15, examining erased images to identify the timestep at which the concept was effectively removed from the majority of images.
> We add more visualizations in the paper in Section A.2 in the Appendix. We demonstrate that focusing exclusively on neurons too early in the denoising trajectory fails to capture all the neurons responsible for generating the target concept. On the other hand, extending beyond 10 timesteps results in a noticeable degradation of image content and quality, striking a delicate balance between effective concept removal and preserving the overall integrity of the image. Therefore, $\hat{t} = 10$ is an optimal point for concept erasure and good retention.
>
>
> [9] Structural Pruning for Diffusion Models, Fang et al.
> [10] The journey, not the destination: How data guides diffusion models, Georgiev, Kristian, et al.
>
> - **Style classifiers instead of CLIP-based metrics** -
>
> We agree with the reviewer that CLIP-based similarity metrics may be considered redundant factors. Initially, we considered utilizing style classifiers, such as those employed in UnlearnDiffAtk or UnlearnCanvas. However, we found that these classifiers often exhibited biases toward specific styles. For instance, the style classifier in UnlearnDiffAtk demonstrated higher accuracy in identifying "Van Gogh" styles but struggled with accurately classifying "Pablo Picasso" styles. As a result, we opted to rely on CLIP similarities, as they are consistently reported across the related works and baselines considered in our study.
>
> **We have addressed all the comments and concerns raised by the reviewer. We sincerely hope that you consider revising the score accordingly.**

---

> ### Comment · Reviewer_B4VY · 2024-11-27
>
> Thank you to the authors for their thoughtful responses and the revisions made to the manuscript. I acknowledge the substantial effort invested in addressing the feedback. Although it is uncommon for significant modifications to occur during the rebuttal phase, given the current circumstances, this is understood as part of the process. I believe that most of my concerns have been adequately addressed, and the authors have effectively defended the advantages and uniqueness of their proposed method. Additionally, the comprehensive survey and elaboration on recent unlearning methods are commendable. Consequently, I am willing to increase my score to a 5, indicating a borderline reject.
>
> However, a key concern remains that prevents me from assigning a more favorable rating: the practicality of the authors' definition of "light-weight." While ConceptPrune avoids the need for backpropagation during model training, determining how to prune the corresponding weights still requires forward inference. Specifically, evaluating the final 10 timesteps is approximately equivalent to performing 10 epochs of inference on the data to be forgotten. Although backpropagation is avoided, the computational cost of 10 epochs of inference is still significant. If the authors could provide an equitable comparison of the unlearning's running time (and memory cost), encompassing both inference and training (where applicable to methods other than ConceptPrune), this would further emphasize the lightweight advantage of ConceptPrune. I would be willing to further increase my score upon receiving such clarifications.
>
> I look forward to the authors' response.

---

> > ### Author Response · Authors · 2024-11-28
> > **Author Response**
> >
> > We thank the reviewer for their prompt feedback. The reviewer correctly notes that ConceptPrune does not require backpropagation but does involve forward passes over the "forgetting data." ***However, we believe there may be a misunderstanding in the statement that "the final 10 timesteps are approximately equivalent to performing 10 epochs of inference on the data to be forgotten."*** While it is true that backpropagation is avoided, equating 10 diffusion timesteps to 10 epochs of inference significantly overstates the computational cost, which remains far lower in our approach.
> >
> > First, we would like to summarize the procedure as follows:
> >
> > We begin by compiling the names of N = 20 common objects (detailed in Table 9 in the Appendix) and construct two types of prompts: concept prompts (e.g., "a cat in the style of Van Gogh") and reference prompts (e.g., "a cat"). The forgetting dataset is composed solely of these prompts. ConceptPrune operates in three main steps:
> > 1. Collecting neuron activations for the first 10 timesteps through forward passes.
> > 2. Calculating WANDA scores and comparisons based on the collected activations.
> > 3. Aggregating scores across timesteps to identify relevant neurons.
> >
> > Since steps (2) and (3) are negligible, the primary cost of ConceptPrune lies in step (1), where neuron activations are gathered through forward passes over the forgetting data.
> >
> > We now compare the computational cost of performing 10 epochs of inference on the forgetting data versus erasing a concept using ConceptPrune. Let D represent the cost of decoding the latents to the pixel space and U represent the cost of running a single denoiser pass. Like other methods, sampling in SD is performed over 50 timesteps.
> > 1. The computational cost for 10 epochs of inference would be - ((50×U)+D)×N×10
> > 2. In contrast, the computational cost for collecting neuron activations for N prompts over only the first 10 timesteps is significantly lower - (10×U)×N
> >
> > Therefore, the computational cost of performing concept erasure using ConceptPrune is much less than 10 epochs of inference on the forgetting data. However, we would like to emphasize that inference for all unlearning methods in our study is performed over 50 steps, ensuring that the inference time remains consistent across all baselines considered in the paper.
> >
> > We provide the time to unlearn a concept using ConceptPrune on a single V100 GPU. From the table below, we can see that ConceptPrune can be considered lightweight compared to methods like CA, ESD, SH, and AdvUnlearn as all of them require backpropagation through the denoiser for multiple epochs.
> >
> > | **Method**        | **Unlearning Time (in secs)** |
> > |--------------------|------------------------------|
> > | ConceptPrune       | **150.1 (2.5 mins)**                  |
> > | CA                 | 762.4  (13 mins)                |
> > | ESD                | 9000 (2.5 hours)           |
> > | SH                 | 4800 (1.3 hours)           |
> > | AdvUnlearn         | 31200 (8.7 hours)          |
> >
> > **We hope we have effectively addressed the reviewer’s concerns and provided satisfactory responses to their questions. We kindly request the reviewer to consider revising their score in favor of recommending acceptance.**

---

> ### Comment · Reviewer_B4VY · 2024-12-01
>
> Thanks for the detailed clarifications. I've correspondingly increased my score to 6 - borderline accept. I encourage the authors to incorporate the updates discussed during the review process into the manuscript.
>
> Furthermore, I recommend that the authors revisit the use of classifier-based evaluation for style removal, as Weakness 5 mentioned. Specifically, the manuscript could benefit from including both CLIP-score-based metrics and classifier-based metrics, accompanied by an in-depth discussion of the results. Such additions would provide a more comprehensive understanding of the proposed approach and further strengthen the manuscript.

---

### Public Comment · ~Finn_Carter1 · 2024-11-22
**Lack of recent related works**

It seems that several recent related works [1,2,3] are ignored.

[1] One-dimensional Adapter to Rule Them All: Concepts, Diffusion Models and Erasing Applications

[2] MACE: Mass Concept Erasure in Diffusion Models

[3] Receler: Reliable Concept Erasing of Text-to-Image Diffusion Models via Lightweight Erasers

---

### Author Response · Authors · 2024-11-27
**Common response**

We thank all the reviewers for their insightful and positive feedback. We are pleased that they found our paper reasonable and acknowledged that our contributions extend beyond concept editing, including training-free erasure and enhanced adversarial robustness. We have revised the paper accordingly with changes marked in blue. We address reviewer comments below and incorporate all feedback.

**Comparison to more baselines (B4VY, PDbh, YHCm, uDZE, and public comment by Finn Carter)**


We appreciate the reviewers’ suggestions regarding additional comparison to recent state-of-the-art (SOTA) methods. In response, we have incorporated the following additional baselines into the paper – Selective Amnesia (SA) [1], SPM [2], Receler [3], MACE [4], Scissorhands[5], AdvUnlearn [6], and SalUn[8], extending the comparison to a total of 11 SOTA methods. However, we would like to argue that some of these are not direct competitors to ConceptPrune.

While the baselines considered in the paper are highly effective, deploying current state-of-the-art concept erasure techniques in real-world scenarios poses significant challenges, particularly in online environments with computational constraints where harmful concepts can emerge dynamically. This is because these methods struggle to meet the following requirements for real-world applications: (1) training-free concept erasure, eliminating concepts without the need for backpropagation through the entire model, or (2) lightweight or fast concept erasure, allowing concepts to be removed quickly and efficiently with minimal compute.

In Table A, we present a comprehensive summary of related works, categorizing them based on whether they are training-free and lightweight for an online setting. For instance, FMN and UCE are considered lightweight as they can achieve concept erasure in approximately 35 and 120 seconds. In contrast, ESD is significantly more time-intensive, making it less practical for rapid applications.

Our proposed solution, ConceptPrune, excels on both fronts by introducing a training-free, pruning-based approach that eliminates harmful concepts without updating any parameters. Instead, it identifies and targets the neurons responsible for generating these concepts enabling efficient concept erasure (in approximately 150 seconds) with significantly reduced computational requirements. Therefore, we consider the methods categorized as lightweight concept erasers to be direct competitors to ConceptPrune. More details in Section 2 (Related Work)

| **Method**          | **Training-free** | **Parameters Trained**       | **Lightweight Erasure** |
|----------------------|-------------------|------------------------------|--------------------------|
| CA            | N                 | Full denoiser                | N                        |
| SA [1]              | N                 | Full denoiser                | N                        |
| SH [5]                | N                 | Full denoiser                | N                        |
| AdvUnlearn [6]     | N                 | Full denoiser                | N                        |
| SalUn [8]            | N                 | Full denoiser                | N                        |
| ESD         | N                 | Cross Attention              | N                        |
| Receler [3]        | N                 | Cross Attention              | Y                        |
| FMN            | N                 | Cross Attention              | Y                        |
| SPM [2]              | N                 | LORA                        | Y                        |
| MACE [4]              | N                 | Cross Attention + LORA       | Y                        |
| UCE     | Y                 | Cross Attention              | Y                        |
| **Ours (ConceptPrune)**             | **Y**             | **_None_**                   | **Y**                    |

**Table A - Summary of recent Concept Erasure baselines. ConceptPrune is a training-free approach that enables rapid pruning of the model to eliminate a new target concept without the need for extensive re-training.**


[1] Selective Amnesia: A Continual Learning Approach to Forgetting in Deep Generative Models
[2] One-dimensional Adapter to Rule Them All: Concepts, Diffusion Models and Erasing Applications, Lyu et al
[3] Receler: Reliable Concept Erasing of Text-to-Image Diffusion Models via Lightweight Erasers, Huang et al
[4] MACE: Mass Concept Erasure in Diffusion Models, Lu et al
[5] Scissorhands: Scrub Data Influence via Connection Sensitivity in Networks, Wu et al
[6] Defensive Unlearning with Adversarial Training for Robust Concept Erasure in Diffusion Models, Zhang et al
[7] Pruning for Robust Concept Erasing in Diffusion Models, Yang et al
[8] Salun: Empowering machine unlearning via gradient-based weight saliency in both image classification and generation. Fan, Chongyu, et al.

---

### Meta-Review · Area_Chair_1uE6 · 2024-12-18

**Metareview:**

The authors provided a detailed response that most reviewers found useful, as it partially or mostly addressed their concerns. Reviewer uDZE, who gave the lowest rating, did not actively participate in the discussion phase. The remaining reviewers maintained a positive stance on this submission. I agree that the paper's strengths significantly outweigh its weaknesses, and I therefore recommend acceptance.

**Additional Comments On Reviewer Discussion:**

The authors did a commendable job in their rebuttal, addressing most of the reviewers' concerns, e.g., on related work and baselines. This was acknowledged by the majority of reviewers during the discussion phase. Therefore, I recommend acceptance.

---

### Decision · Program_Chairs · 2025-01-22

Accept (Poster)